# Bandits with Knapsacks beyond the Worst Case

**Karthik Abinav Sankararaman**
Facebook AI, Menlo Park
karthikabinavs@gmail.com

**Aleksandrs Slivkins**
Microsoft Research, New York City
slivkins@microsoft.com

## Abstract

Bandits with Knapsacks (BwK) is a general model for multi-armed bandits under supply/budget constraints. While worst-case regret bounds for BwK are well-understood, we present three results that go beyond the worst-case perspective. First, we provide upper and lower bounds which amount to a *full characterization* for logarithmic, instance-dependent regret rates. Second, we consider "simple regret" in BwK, which tracks algorithm's performance in a given round, and prove that it is small in all but a few rounds. Third, we provide a general "reduction" from BwK to bandits which takes advantage of some known helpful structure, and apply this reduction to combinatorial semi-bandits, linear contextual bandits, and multinomial-logit bandits. Our results build on the BwK algorithm from Agrawal and Devanur [3], providing new analyses thereof.

## 1 Introduction

We study multi-armed bandit problems with supply or budget constraints. Multi-armed bandits is a simple model for *exploration-exploitation tradeoff*, *i.e.,* the tension between acquiring new information and making optimal decisions. It is an active research area, spanning computer science, operations research, and economics. Supply/budget constraints arise in many realistic applications, *e.g.,* a seller who dynamically adjusts the prices or product assortment may have a limited inventory, and an algorithm that optimizes ad placement is constrained by the advertisers' budgets. Other motivating examples concern repeated auctions, crowdsourcing markets, and network routing.

We consider a general model called *Bandits with Knapsacks* (*BwK*), which subsumes the examples mentioned above. There are $d \geq 2$ *resources* that are consumed over time, one of which is time itself. Each resource $i$ starts out with budget $B_i$. In each round $t$, the algorithm chooses an action (*arm*) $a = a_t$ from a fixed set of $K$ actions. The outcome is a vector in $[0,1]^{d+1}$: it consists of a reward and consumption of each resource. This vector is drawn independently from some distribution over $[0,1]^{d+1}$, which depends on the chosen arm but not on the round, and is not known to the algorithm. The algorithm observes *bandit feedback*, *i.e.,* only the outcome of the chosen arm. The algorithm stops at a known time horizon $T$, or when the total consumption of some resource exceeds its budget. The goal is to maximize the total reward, denoted REW.

The presence of supply/budget constraints makes the problem much more challenging. First, algorithm's choices constrain what it can do in the future. Second, the algorithm is no longer looking for arms with maximal expected per-round reward (because such arms may consume too much resources). Third, the best fixed distribution over arms can be much better than the best fixed arm. Accordingly, we compete with the *best fixed distribution* benchmark: the total expected reward of the best distribution, denoted $\mathtt{OPT_{FD}}$. All this complexity is already present even when $d = 2$, *i.e.,* when there is only one resource other than time, and the minimal budget is $B = \min_i B_i = \Omega(T)$.

BwK were introduced in [14, 16] and extensively studied since then. The optimal worst-case regret rate is well-understood. In particular, it is $\tilde{\mathcal{O}}(\sqrt{KT})$ when $B = \Omega(T)$.

35th Conference on Neural Information Processing Systems (NeurIPS 2021).

We present several results that go beyond the worst-case perspective:

**1.** We provide a full characterization for instance-dependent regret rates. In stochastic bandits, one obtains regret $\mathcal{O}\left(\frac{K}{\Delta}\log T\right)$, where $\Delta$ is the the *reward-gap*: the gap in expected reward between the best and the second-best arm. We work out whether, when and how such results extend to `BwK`.

**2.** We show that *simple regret*, which tracks algorithm's performance in a given round, can be small in all but a few rounds. Like in stochastic bandits, simple regret can be at least $\epsilon$ in at most $\tilde{\mathcal{O}}(K/\epsilon^2)$ rounds, and this is achieved for all $\epsilon > 0$ simultaneously.

**3.** We improve all results mentioned above for a large number of arms, assuming some helpful structure. In fact, we provide a general "reduction" from `BwK` to stochastic bandits, and apply this reduction to three well-studied scenarios from stochastic bandits.

Our algorithmic results focus on `UcbBwK`, a `BwK` algorithm from [3] which implements the "optimism under uncertainty" paradigm and attains the optimal worst-case regret bound. We provide new analyses of this algorithm along the above-mentioned themes.

**Related work.** Background on multi-armed bandits can be found in books [23, 54, 42]. *Stochastic bandits (i.e.,* `BwK` without resources) is a basic, well-understood version. The dependence on $\Delta$ and $\epsilon$ are optimal as stated above [41, 10, 11], and is achieved simultaneously with the optimal worst-case regret $\widetilde{O}(KT)$, *e.g.,* in [10]. Various refinements are known for $O(\log T)$ regret [10, 8, 34, 32, 45]. Most relevant to this paper is $\mathcal{O}\left(\sum_a \log(T)/\Delta(a)\right)$ regret, where $\Delta(a)$ is the gap in expected reward between arm $a$ and the best arm [10]. Improving regret for large / infinite number of arms via a helpful structure is a unifying theme for several prominent lines of work, *e.g.,* linear bandits, convex bandits, Lipschitz bandits, and combinatorial (semi-)bandits.

Bandits with Knapsacks were introduced in [14, 16], and optimally solved in the worst case. Subsequent work extended `BwK` to a more general notion of rewards/consumptions [3], combinatorial semi-bandits [49], and contextual bandits [15, 6, 4]. Several special cases with budget/supply constraints were studied separately (and inspired a generalization to `BwK`): dynamic pricing [19, 12, 20, 59], dynamic procurement [13, 52], and dynamic ad allocation [53, 28]. The adversarial version of `BwK` was studied by [35, 36]. All this work considers worst-case regret bounds.

Several papers achieve $O(\log T)$ regret in `BwK`, but with substantial caveats that we avoid. [61] assume deterministic consumption, whereas all motivating examples of `BwK` require stochastic consumption correlated with rewards (*e.g.,* dynamic pricing consumes supply only if a sale happens). They posit $d = 2$ and no other assumptions, whereas we show that "best-arm optimality" is necessary with stochastic consumption. [31] assume "best-arm-optimality" as we do (it is implicit in their version of reward-gap). However, their algorithm inputs an instance-dependent parameter which is "hidden" in `BwK`. Moreover, their $O(\log T)$ regret bound scales with $c_{\min}$, minimal expected consumption among arms (as $c_{\min}^{-4}$). Their worst-case regret bound is suboptimal, since it also scales with $c_{\min}$ (as $c_{\min}^{-2}$), and only applies for $d = 2$. [58] study a contextual version of `BwK` with two arms, one of which does nothing; this is meaningless when specialized to `BwK`. [44], subsequent to our initial draft on `arxiv.org`, use extra parameters (other than a version of reward-gap), which yield $\geq \sqrt{T}$ regret whenever our lower bounds apply;[1] it is unclear when all their parameters are small. No worst-case regret bounds are provided; their algorithm does not appear to achieve even $o(T)$ regret in the worst case. Finally, [33, 56, 57, 30, 47] posit one constrained resource and $T = \infty$. This is an easier problem, *e.g.,* the best arm is the best distribution over arms.

## 2 Preliminaries: the problem, linear relaxation and `UcbBwK` algorithm

The bandits with knapsacks (`BwK`) problem is as follows. There are $K$ arms, $d$ resources, and $T$ rounds. Initially, each resource $j \in [d]$ is endowed with budget $B_j$. In each round $t = 1, \ldots, T$, an algorithm chooses an arm $a_t$, and observes an outcome vector $\boldsymbol{o}_t = (r_t; c_{1,t}, \ldots, c_{d,t}) \in [0,1]^{d+1}$, where $r_t$ is the reward, and $c_{j,t}$ is the consumption of each resource $j$. The algorithm stops when the consumption of some resource $j$ exceeds its budget $B_j$, or after $T$ rounds, whichever is sooner. We maximize the total reward, `REW` $= \sum_{t=1}^{\tau} r_t$, where $\tau$ is the stopping time. We focus on the stochastic version: for each arm $a$, there is a distribution $\mathcal{D}_a$ over $[0,1]^{d+1}$ such that each outcome vector $\boldsymbol{o}_t$ is

---

[1]Conceptually, our assumption of "best-arm-optimality" is replaced with another assumption: a lower bound on the positive entries of the optimal distribution $x^*$ (parameter $\chi$ in Section 3.3 of [44]).

an independent draw from distribution $\mathcal{D}_{a_t}$ (which depends only on the chosen arm $a_t$). A problem instance consists of parameters $(K, d, T;\ B_1,\ \ldots,\ B_d)$ and distributions $(\mathcal{D}_a : \text{arms } a)$.

Given a problem instance, the *best dynamic policy* benchmark $\mathtt{OPT_{DP}}$ maximizes the total expected reward over all algorithms; it is used in all worst-case regret bounds. The *best fixed distribution* benchmark $\mathtt{OPT_{FD}}$, used in some of our results, maximizes the total expected reward over all algorithms that always sample an arm from the same distribution. The worst-case optimal regret rate is [16]:

$$\mathtt{OPT_{DP}} - \mathbb{E}[\mathtt{REW}] = \tilde{\mathcal{O}}(\ \sqrt{K\,\mathtt{OPT_{DP}}} + \mathtt{OPT_{DP}}\sqrt{K/B}\ ), \quad B = \min_{j \in [d]} B_j. \tag{2.1}$$

**Simplifications and notation.** Following prior work, we make three assumptions without losing generality. First, all budgets are the same: $B_1 = \ldots = B_d = B$. This is w.l.o.g. because one can divide the consumption of each resource $j$ by $B_j / \min_i B_i$; dependence on the budgets is driven by the smallest $B_j$. Second, resource $d$ corresponds to time: each arm deterministically consumes $B/T$ units of this resource in each round. It is called the *time resource* and denoted $\mathtt{time}$. Third, there is a *null arm*, denoted $\mathtt{null}$, whose reward and consumption of all resources except $\mathtt{time}$ is always $0$.[2]

Like most prior work on $\mathtt{BwK}$, we use $\mathcal{O}(\cdot)$ notation rather than track explicit constants in regret bounds. This improves clarity and emphasizes the more essential aspects of analyses and results.

For $n \in \mathbb{N}$, let $[n] = \{1,\ \ldots,\ n\}$ and $\Delta_n = \{\text{all distributions on } [n]\}$. Let $[K]$ and $[d]$ be, resp., the set of all arms and the set of all resources. For each arm $a$, let $r(a)$ and $c_j(a)$ be, resp., the mean reward and mean resource-$j$ consumption, *i.e.*, $(r(a); c_1(a),\ \ldots,\ c_d(a)) := \mathbb{E}_{\boldsymbol{o} \sim \mathcal{D}_a}[\boldsymbol{o}]$. We sometimes write $\boldsymbol{r} = (r(a) : a \in [K])$ and $\boldsymbol{c}_j = (c_j(a) : a \in [K])$ as vectors over arms. Given a function $f : [K] \to \mathbb{R}$, we extend it to distributions $\boldsymbol{X}$ over arms as $f(\boldsymbol{X}) := \mathbb{E}_{a \sim \boldsymbol{X}}[f(a)]$.

**Linear Relaxation.** Following prior work, we consider a linear relaxation:

$$\begin{aligned}
\text{maximize} \quad & \boldsymbol{X} \cdot \boldsymbol{r} && \text{such that} \\
& \boldsymbol{X} \in [0,1]^K,\ \boldsymbol{X} \cdot \boldsymbol{1} = 1 \\
\forall j \in [d] \quad & \boldsymbol{X} \cdot \boldsymbol{c}_j \le B/T.
\end{aligned} \tag{2.2}$$

Here $\boldsymbol{X}$ is a distributions over arms, the algorithm does not run out of resources in expectation, and the objective is the expected per-round reward. Let $\mathtt{OPT_{LP}}$ be the value of this linear program. Then $\mathtt{OPT_{LP}} \ge \mathtt{OPT_{DP}}/T \ge \mathtt{OPT_{FD}}/T$ [16]. The Lagrange function $\mathcal{L} : \Delta_K \times \mathbb{R}_+^d \to \mathbb{R}$ defined as follows:

$$\mathcal{L}(\boldsymbol{X}, \boldsymbol{\lambda}) := r(\boldsymbol{X}) + \sum_{j \in [d]} \lambda_j [\ 1 - {}^T\!/_B\, c_j(\boldsymbol{X}),\ ]. \tag{2.3}$$

where $\boldsymbol{\lambda}$ corresponds to the dual variables. Then (*e.g.*, by Theorem D.2.2 in [17]):

$$\min_{\boldsymbol{\lambda} \ge 0} \max_{\boldsymbol{X} \in \Delta_K} \mathcal{L}(\boldsymbol{X}, \boldsymbol{\lambda}) = \max_{\boldsymbol{X} \in \Delta_K} \min_{\boldsymbol{\lambda} \ge 0} \mathcal{L}(\boldsymbol{X}, \boldsymbol{\lambda}) = \mathtt{OPT_{LP}}. \tag{2.4}$$

The $\min$ and $\max$ in (2.4) are attained, so that $(\boldsymbol{X}^*, \boldsymbol{\lambda}^*)$ is maximin pair if and only if it is minimax pair; such pair is called a *saddle point*. We'll use $\mathcal{L}(\,\cdot\,, \boldsymbol{\lambda}^*)$ to generalize reward-gap to $\mathtt{BwK}$.

**Algorithm $\mathtt{UcbBwK}$.** We analyze an algorithm from [3], defined as follows. In the LP (2.2), rescale the last constraint, for each resource $j \ne \mathtt{time}$, as $(^B\!/_T)(1 - \eta_{\mathrm{LP}})$, where

$$\eta_{\mathrm{LP}} := 3 \cdot (\ \sqrt{K/B\ \log(KdT)} + {}^K\!/_B\,(\log(KdT))^2\ ). \tag{2.5}$$

We call it the *rescaled LP* (see (C.1)). Its value is $(1 - \eta_{\mathrm{LP}})\,\mathtt{OPT_{LP}}$. At each round $t$, the algorithm forms an "optimistic" version of this LP, upper-bounding rewards and lower-bounding consumption:

$$\begin{aligned}
\text{maximize} \quad & \sum_{a \in [K]} X(a)\, r_t^+(a) && \text{such that} \\
& \boldsymbol{X} \in [0,1]^K,\ \sum_{a \in [K]} X(a) = 1 \\
\forall j \in [d] \quad & \sum_{a \in [K]} X(a)\, c_{j,t}^-(a) \le B(1 - \eta_{\mathrm{LP}})/T.
\end{aligned} \tag{2.6}$$

$\mathtt{UcbBwK}$ solves (2.6), obtains distribution $\boldsymbol{X}_t$, and samples an arm $a_t$ independently from $\boldsymbol{X}_t$. The algorithm achieves the worst-case optimal regret bound in (2.1). The upper/lower confidence bounds $r_t^+(a),\ c_{j,t}^-(a) \in [0,1]$ are computed in a particular way specified in Appendix B. What matters to this paper is that they satisfy a high-probability event

$$0 \le r_t^+(a) - r(a) \le \mathrm{Rad}_t(a) \text{ and } 0 \le c_j(a) - c_{j,t}^-(a) \le \mathrm{Rad}_t(a), \tag{2.7}$$

---

[2]Choosing the null arm is equivalent to skipping a round. One can take an algorithm $\mathtt{ALG}$ that uses $\mathtt{null}$, and turn it into an algorithm that doesn't: when $\mathtt{ALG}$ chooses $\mathtt{null}$, just call it again until it doesn't.

for some *confidence radius* $\text{Rad}_t(a)$ specified below. This event holds, simultaneously for all arms $a$, resources $j$ and rounds $t$, with probability (say) at least $1 - \frac{\log(KdT)}{T^4}$. For $a \neq \texttt{null}$, we can take

$$\text{Rad}_t(a) = \min(\ 1,\ \sqrt{C_{\text{rad}}/N_t(a)} + C_{\text{rad}}/N_t(a)\ ), \tag{2.8}$$

where $C_{\text{rad}} = 3 \cdot \log(KdT)$ and $N_t(a)$ is the number of rounds before $t$ in which arm $a$ has been chosen. There is no uncertainty on the time resource and the null arm, so we define $c^-_{\texttt{time},\,t}(\cdot) = B/T$ and $\text{Rad}_t(\texttt{null}) = r^+_t(\texttt{null}) = c^-_{j,t}(\texttt{null}) = 0$ for all resources $j \neq \texttt{time}$.

# 3 Logarithmic instance-dependent regret bounds

We provide upper and lower bounds which amount to *full characterization* of logarithmic, instance-dependent regret rates in BwK. We achieve $O(\log T)$ regret under two assumptions: there is only one resource other than time (*i.e.,* $d = 2$), and the best distribution over arms reduces to the best fixed arm (*best-arm-optimality*). We prove that both assumptions are essentially necessary for any algorithm, deriving complementary $\Omega(\sqrt{T})$ lower bounds if either assumption fails. Both lower bounds hold in a wide range of problem instances; arguably, they represent typical scenarios rather than exceptions. All upper and lower bounds are against the best fixed distribution benchmark ($\texttt{OPT}_{\texttt{FD}}$).

We achieve $O(\log T)$ regret with UcbBwK algorithm [3], which implies two very desirable properties: the algorithm does not know in advance whether best-arm-optimality holds, and attains the optimal worst-case regret bound for all instances, best-arm-optimal or not. The positive result would have been weaker without either property, although still non-trivial.

We identify a suitable instance-dependent parameter, defined via Lagrangians from Eq. (2.3):

$$G_{\texttt{LAG}}(a) := \texttt{OPT}_{\texttt{LP}} - \mathcal{L}(a, \boldsymbol{\lambda}^*) \qquad \textit{(Lagrangian gap of arm a)}, \tag{3.1}$$

where $\boldsymbol{\lambda}^*$ is a minimizer in Eq. (2.4). It is a non-obvious generalization of the *reward-gap* from multi-armed bandits, $\Delta(a) = \max_{a'} r(a') - r(a)$. The Lagrangian gap of a problem instance is

$$G_{\texttt{LAG}} := \min_{a \notin \{a^*, \texttt{null}\}} G_{\texttt{LAG}}(a). \tag{3.2}$$

Our regret bound scales as $\mathcal{O}(KG_{\texttt{LAG}}^{-1} \log T)$, which is optimal in $G_{\texttt{LAG}}$, under a mild additional assumption, and as $\mathcal{O}(KG_{\texttt{LAG}}^{-2} \log T)$ otherwise.

## 3.1 $O(\log T)$ regret analysis for UcbBwK

We analyze a version of UcbBwK which "prunes out" the null arm, call it PrunedUcbBwK. (This modification can only improve regret, so it retains the worst-case regret (2.1) of UcbBwK.) We provide a new analysis of this algorithm for $d = 2$ and best-arm-optimality. We analyze the sensitivity of the "optimistic" linear relaxation to small perturbations in the coefficients, and prove that the best arm is chosen in all but a few rounds. The key is to connect each arm's confidence term with its Lagrangian gap. This gives us $\mathcal{O}(KG_{\texttt{LAG}}^{-2} \log T)$ regret rate. To improve it to $\mathcal{O}(KG_{\texttt{LAG}}^{-1} \log T)$, we use a careful counting argument which accounts for rewards and consumption of non-optimal arms.

Algorithm PrunedUcbBwK is formally defined as follows: in each round $t$, call UcbBwK as an oracle, repeat until it chooses a non-null arm $a$, and set $a_t = a$. (In one "oracle call", UcbBwK outputs an arm and inputs an outcome vector for this arm.) The total number of oracle calls is capped at $N_{\max} = \alpha_0 \cdot T^2 \, \log T$, with a sufficiently large absolute constant $\alpha_0$ which we specify later in Claim 3.6. Formally, after this many oracle calls the algorithm can only choose the null arm.

**Definition 3.1.** *An instance of BwK is called* best-arm-optimal *with best arm $a^* \in [K]$ if the following conditions hold: (i) $\texttt{OPT}_{LP} = \frac{B}{T} \cdot r(a^*) / \max_{j \in [d]} c_j(a^*)$, (ii) the linear program (2.2) has a unique optimal solution $\boldsymbol{X}^*$ supported on $\{a^*, \texttt{null}\}$, and (iii) $X^*(a^*) > \frac{3\sqrt{B} \log(KdT)}{T}$.*

Part (ii) here is essentially w.l.o.g.;[3] part (iii) states that the optimal value should not be tiny.

---

[3] Part (ii) holds almost surely given part (i) if one adds a tiny noise, *e.g.,* $\epsilon$-variance, mean-0 Gaussian for any $\epsilon > 0$, independently to each coefficient in the LP (2.2), as per Prop. 3.1 in [46]. To implement this, an algorithm can precompute the noise terms and add them consistently to observed rewards and consumptions.

We assume $d = 2$ and best-arm-optimality throughout this section without further mention. In particular, the linear program (2.2) has a unique optimal solution $\boldsymbol{X}^*$, and its support has only one arm $a^* \neq \texttt{null}$. We use $c(a)$ to denote the mean consumption of the non-time resource on arm $a$. We distinguish two cases, depending on whether $c(a^*)$ is very close to $B/T$.

**Theorem 3.2.** *Fix a best-arm optimal problem instance with only one resource other than time* (i.e., $d = 2$). *Consider Algorithm* `PrunedUcbBwK` *with parameter* $\eta_{\text{LP}} \leq \frac{1}{2}$ *in* (2.5). *Then*

    (i) *$OPT_{FD} - \mathbb{E}[REW] \leq \mathcal{O}\left(\frac{OPT_{FD}}{B} \cdot \Psi\right)$, where $\Psi := \sum_{a \notin \{a^*, \texttt{null}\}} G_{LAG}^{-2}(a) \cdot \log(KdT)$.*

    (ii) *Moreover, if $|c(a^*) - B/T| > \Omega(\Psi/T)$, then*

$$OPT_{FD} - \mathbb{E}[REW] \leq \mathcal{O}\left( \sum_{a \notin \{a^*, \texttt{null}\}} G_{LAG}^{-1}(a) \, \log(KdT) \right). \tag{3.3}$$

Eq. (3.3) optimally depends on $G_{\text{LAG}}(\cdot)$: indeed, it does in the unconstrained case when Lagrangian gap specializes to the reward gap, as per the lower bound in [41]. In particular, Eq. (3.3) holds if $G_{\text{LAG}} > T^{-1/4}$ and $|c(a^*) - B/T| > \mathcal{O}(T^{-1/2})$. The constant in $\mathcal{O}(\cdot)$ is $48$ in both parts of the theorem; the analysis only suppresses constants from concentration bounds and from Lemma 3.3.

### 3.1.1 Basic analysis: proof of Theorem 3.2(i)

We analyze `UcbBwK` in a relaxed version of `BwK`, where an algorithm runs for exactly $N_{\max}$ rounds, regardless of the time horizon and the resource consumption; call it *Relaxed BwK*. The algorithms are still parameterized by the original $B, T$, and observe the resource consumption.

We sometimes condition on the high-probability event that (2.7) holds for all rounds $t \in [N_{\max}]$, call it the "clean event". Recall that its probability is at least $1 - \frac{\mathcal{O}(\log(KdT))}{T^2}$.

We prove that the best arm $a^*$ chosen in all but a few rounds. The crux is an argument about sensitivity of linear programs to perturbations. More specifically, we argue about sensitivity of the support of the optimal solution for the linear relaxation (2.2).

**Lemma 3.3** (LP-sensitivity). *Consider an execution of* `UcbBwK` *in Relaxed BwK. Under the "clean event", $\text{Rad}_t(a) \geq \frac{1}{4} G_{LAG}(a)$ for each round $t$ and each arm $a \in \text{supp}(\boldsymbol{X}_t) \setminus \{a^*, \texttt{null}\}$.*

**Proof Sketch** We use a standard result about LP-sensitivity, the details are spelled out in Appendix C. We apply this result via the following considerations. We treat the optimistic LP (2.6) a perturbation of (the rescaled version of) the original LP (2.2). We rely on perturbations being "optimistic" (*i.e.,* upper-bounding rewards and lower-bounding resource consumption). We use the clean event to upper-bound the perturbation size by the confidence radius. Finally, we prove that

$$G_{\text{LAG}}(a) = \frac{T}{B} \sum_{j \in [d]} \lambda_j^* c_j(a) - r(a), \tag{3.4}$$

and use this characterization to connect Lagrangian gap to the allowed perturbation size. ∎

We rely on the following fact which easily follows from the definition of the confidence radius:

**Claim 3.4.** *Consider an execution of some algorithm in Relaxed BwK. Fix a threshold $\theta > 0$. Then each arm $a \neq \texttt{null}$ can only be chosen in at most $\mathcal{O}\left(\theta^{-2} \log(KdT)\right)$ rounds $t$ with $\text{Rad}_t(a) \geq \theta$.*

**Corollary 3.5.** *Consider an execution of* `UcbBwK` *in Relaxed BwK. Under the clean event, each arm $a \notin \{a^*, \texttt{null}\}$ is chosen in at most $N_0(a) := \mathcal{O}\left(G_{LAG}^{-2}(a) \, \log(KdT)\right)$ rounds.*

This follows from Lemma 3.3 and Claim 3.4. Next, the null arm is not chosen too often:

**Claim 3.6.** *Consider an execution of* `UcbBwK` *in Relaxed BwK. With probability at least $1 - \mathcal{O}(T^{-3})$, the following happens: the null arm cannot be chosen in any $\alpha_0 T \log(T)$ consecutive rounds, for a large enough absolute constant $\alpha_0$. Consequently, a non-null arm is chosen in at least $T$ rounds.*

**Proof Sketch** Fix round $t$, and suppose `UcbBwK` chooses the null arm in $N$ consecutive rounds, starting from $t$. No new data is added, so the optimistic LP stays the same throughout. Consequently, the solution $\boldsymbol{X}_t$ stays the same, too. Thus, we have $N$ consecutive independent draws from $\boldsymbol{X}_t$ that return `null`. It follows that $r(\boldsymbol{X}_t) < 1/T$ with high probability, *e.g.,* by (B.2). On the other hand, assume the clean event. Then $r(\boldsymbol{X}_t) \geq (1 - \eta_{\text{LP}}) OPT_{\text{LP}}$ by definition of the optimistic LP, and consequently $r(\boldsymbol{X}_t) \geq (1 - \eta_{\text{LP}}) OPT_{\text{DP}}/T$. We obtain a contradiction. ∎

Corollary 3.5 and Claim 3.6 imply a strong statement about the pruned algorithm.

**Claim 3.7.** *Consider an execution of* `PrunedUcbBwK` *in the (original)* `BwK` *problem. With probability at least* $1 - \mathcal{O}(T^{-2})$, *each arm* $a \notin \{a^*, \texttt{null}\}$ *is chosen in at most* $N_0(a)$ *rounds, and arm* $a^*$ *is chosen in* $T - N_0$ *remaining rounds,* $N_0 := \sum_{a \notin \{a^*, \texttt{null}\}} N_0(a)$.

We take a very pessimistic approach to obtain Theorem 3.2(i): we only rely on rewards collected by arm $a^*$, and we treat suboptimal arms as if they bring no reward and consume the maximal possible amount of resource. We formalize this idea as follows (see Appendix D for details).

For a given arm $a$, let $\texttt{REW}(a)$ be the total reward collected by arm $a$ in `PrunedUcbBwK`. Let $\texttt{REW}(a \mid B_0, T_0)$ be the total reward of an algorithm that always plays arm $a$ if the budget and the time horizon are changed to $B_0 \leq B$ and $T_0 \leq T$, respectively. Note that

$$\texttt{LP}(a \mid B_0, T_0) := \mathbb{E}[\texttt{REW}(a \mid B_0, T_0)] = r(a) \cdot \min\left( T_0, \frac{B_0}{c(a)} \right). \tag{3.5}$$

is the value of always playing arm $a$ in a linear relaxation with the same constraints. By best-arm-optimality, we have $\mathbb{E}[\texttt{REW}(a^* \mid B, T)] = \texttt{OPT}_{\texttt{FD}}$. We observe that

$$\mathbb{E}[\texttt{REW}(a^* \mid B_0, T_0)] \geq \frac{\min\{T_0, B_0\}}{B} \cdot \texttt{OPT}_{\texttt{FD}}. \tag{3.6}$$

By Claim 3.7 there are at least $B_0 = B - N_0$ units of budget and at least $T_0 = T - N_0$ rounds left for arm $a^*$ with high probability. Consequently,

$$\mathbb{E}[\texttt{REW}] \geq \mathbb{E}[\texttt{REW}(a^*)] \geq \mathbb{E}[\texttt{REW}(a^* \mid B_0, T_0)] - \tilde{\mathcal{O}}(1/T). \tag{3.7}$$

We obtain Theorem 3.2(i) by plugging these $B_0, T_0$ into Eq. (3.6), and then using (3.7).

### 3.1.2 Tighter computation: proof of Theorem 3.2(ii)

We re-use the basic analysis via Claim 3.7, but perform the final computation more carefully so as to account for the rewards and resource consumption of the suboptimal arms.

Let's do some prep-work. First, we characterize $\texttt{REW}(a^*)$ in a more efficient way compared to Eq. (3.7). Let $B(a), T(a)$ denote, resp., the budget and time consumed by `PrunedUcbBwK` when playing a given arm $a$. We use expectations of $B(a)$ and $T(a)$, rather than lower bounds:

$$\mathbb{E}[\texttt{REW}(a)] = r(a) \, \mathbb{E}[T(a)] = r(a) \frac{\mathbb{E}[B(a)]}{c(a)}$$
$$= \texttt{LP}\left( a \mid \mathbb{E}[B(a)], \mathbb{E}[T(a)] \right) \qquad \text{for each arm } a. \tag{3.8}$$

We prove Eq. (3.8) via martingale techniques, see Appendix D.5.

Second, we use a tighter version of Eq. (3.6) (see Appendix D.3): for any $B_0 \leq B$, $T_0 \leq T$

$$\texttt{LP}(a^* \mid B_0, T_0)] \geq \texttt{OPT}_{\texttt{FD}} \cdot \frac{B_0}{B} \, / \left( \max\left\{ \frac{B}{T}, c(a^*) \right\} \cdot \max\left\{ \frac{B_0}{T_0}, c(a^*) \right\} \right). \tag{3.9}$$

Third, we lower-bound $G_{\texttt{LAG}}(a)$ in a way that removes Lagrange multipliers $\lambda^*$:

$$G_{\texttt{LAG}}(a) \geq \begin{cases} \texttt{OPT}_{\texttt{FD}}/T - r(a) & \text{if } c(a^*) < B/T, \\ \texttt{OPT}_{\texttt{FD}} \cdot c(a)/B - r(a) & \text{if } c(a^*) > B/T. \end{cases} \tag{3.10}$$

We derive this from Eq. (3.4) and complementary slackness, see Appendix D.4.

Fourth, let $B_0 = \mathbb{E}[B(a^*)]$ and $T_0 = \mathbb{E}[T(a^*)]$ denote, resp., the expected budget and time consumed by arm $a^*$. Let $N(a) = \mathbb{E}[T(a)]$ be the expected number of pulls for each arm $a \notin \{a^*, \texttt{null}\}$. In this notation, Eq. (3.8) implies that

$$\mathbb{E}[\texttt{REW}] = \sum_{a \notin \{a^*, \texttt{null}\}} N(a) \, r(a) + \texttt{LP}(a^* \mid B_0, T_0). \tag{3.11}$$

Now we are ready for the main computation . We consider four cases, depending on how $c(a^*)$ compares with $B/T$ and $B_0/T_0$. We prove the desired regret bound when $c(a^*)$ is either larger than both or smaller than both, and we prove that it cannot lie in between. The "in-between" cases is the only place in the analysis where we use the assumption that $c(a^*)$ is close to $B/T$.

**Case 1:** $c(a^*) < \min(B/T, B_0/T_0)$. Plugging in Eq. (3.9) into Eq. (3.11) and simplifying,

$$\mathbb{E}[\texttt{REW}] \geq \sum_{a \notin \{a^*, \texttt{null}\}} N(a) \, r(a) + \texttt{OPT}_{\texttt{FD}} \cdot T_0/T. \tag{3.12}$$

Re-arranging, plugging in $T_0 = T - \sum_{a \neq a^*} N(a)$ and simplifying, we obtain

$$\mathtt{OPT_{FD}} - \mathbb{E}[\mathtt{REW}] \leq \sum_{a \notin \{a^*, \mathtt{null}\}} N(a) \left( \tfrac{\mathtt{OPT_{FD}}}{T} - r(a) \right) \tag{3.13}$$
$$\leq \sum_{a \notin \{a^*, \mathtt{null}\}} N(a) \, G_{\mathtt{LAG}}(a) \qquad \textit{(by Eq. (3.10))}$$
$$\leq \mathcal{O}\big( \sum_{a \notin \{a^*, \mathtt{null}\}} G_{\mathtt{LAG}}^{-1}(a) \, \log(KdT) \, \big) \qquad \textit{(by Claim 3.7)}.$$

**Case 2:** $c(a^*) > \max(B/T, B_0/T_0)$. Plugging in Eq. (3.9) into Eq. (3.11) and simplifying,

$$\mathbb{E}[\mathtt{REW}] \geq \sum_{a \notin \{a^*, \mathtt{null}\}} N(a) \, r(a) + \mathtt{OPT_{FD}} \cdot B_0/B. \tag{3.14}$$

Re-arranging, plugging in $B_0 = B - \sum_{a \neq a^*} N(a) \, c(a)$, and simplifying, we obtain

$$\mathtt{OPT_{FD}} - \mathbb{E}[\mathtt{REW}] \leq \sum_{a \notin \{a^*, \mathtt{null}\}} N(a) \left( \tfrac{\mathtt{OPT_{FD}}}{B} \cdot c(a) - r(a) \right)$$
$$\leq \sum_{a \notin \{a^*, \mathtt{null}\}} N(a) \, G_{\mathtt{LAG}}(a) \qquad \textit{(by Eq. (3.10))},$$

and we are done by Claim 3.7, just like in Case 1.

**Case 3:** $B_0/T_0 \leq c(a^*) \leq B/T$. Let us write out $B_0$ and $T_0$:

$$c(a^*) \geq \frac{B_0}{T_0} = \frac{B - \sum_{a \notin \{a^*, \mathtt{null}\}} N(a) \, c(a)}{T - \sum_{a \notin \{a^*, \mathtt{null}\}} N(a)} \geq \frac{B}{T} \left( 1 - \frac{1}{B} \cdot \sum_{a \notin \{a^*, \mathtt{null}\}} N(a) \right)$$
$$\geq B/T - O(\Psi/T), \text{ where } \Psi \text{ is as in Theorem 3.2} \qquad \textit{(by Claim 3.7)}.$$

Since $c(a^*) \leq B/T$, we have $0 \leq B/T - c(a^*) \leq O(\Psi/T)$ which contradicts the premise.

**Case 4:** $B/T \leq c(a^*) \leq B_0/T_0$. The argument is similar to Case 3. Writing out $B_0, T_0$, we have

$$c(a^*) \leq \frac{B_0}{T_0} = \frac{B - \sum_{a \notin \{a^*, \mathtt{null}\}} N(a) c(a)}{T - \sum_{a \notin \{a^*, \mathtt{null}\}} N(a)} \leq \frac{B}{T(1 - \frac{1}{T} \cdot \sum_{a \notin \{a^*, \mathtt{null}\}} N(a))}.$$

By Claim 3.7, $c(a^*) \leq B/T \, (1 + O(\Psi/T))$. Therefore, $0 \leq c(a^*) - B/T \leq O(\Psi/T)$, contradiction.

## 3.2 Lower Bounds (for arbitrary algorithms)

We provide two lower bounds to complement Theorem 3.2: we argue that regret $\Omega(\sqrt{T})$ is essentially inevitable if a problem instance is far from best-arm-optimal or if there are $d > 2$ resources.

We consider problem instances with three arms $\{A_1, A_2, \mathtt{null}\}$, Bernoulli rewards, and $d \geq 2$ resources, one of which is time; call them $3 \times d$ *instances*. Each lower bound constructs two similar problem instances $\mathcal{I}, \mathcal{I}'$ such that any algorithm incurs high regret on at least one of them.[4] The two instances have the same parameters $T, K, d, B$, and the mean reward and the mean consumption for each arm and each resource differ by at most $\epsilon$; we call them $\epsilon$-*perturbation* of each other.

We start with an "original" problem instance $\mathcal{I}_0$ and construct problem instances $\mathcal{I}, \mathcal{I}'$ that are small perturbations of $\mathcal{I}_0$. This is a fairly general result: unlike many bandit lower bounds that focus on a specific pair $\mathcal{I}, \mathcal{I}'$, we allow a wide range for $\mathcal{I}_0$, as per the assumption below.

**Assumption 3.8.** *There exists an absolute constant $c_{\mathtt{LB}} \in (0, 1/3)$ such that:*

1. $r(A_i), c_j(A_i) \in [c_{\mathtt{LB}}, 1 - c_{\mathtt{LB}}]$ *for each arm $i \in \{1, 2\}$ and each resource $j$.*
2. $r(A_2) - r(A_1) \geq c_{\mathtt{LB}}$ *and* $c_j(A_2) - c_j(A_1) \geq c_{\mathtt{LB}} + G_{\mathcal{LAG}}$ *for every resource $j \in [d]$.*
3. $B \leq c_{\mathtt{LB}} \cdot T \leq \mathtt{OPT_{FD}}$.
4. *Lagrangian gap is not extremely small: $G_{\mathcal{LAG}} \geq c_{\mathtt{LB}}/\sqrt{T}$.*

For a concrete example, let us construct a family of $3 \times d$ problem instances that satisfy these assumptions. Fix some absolute constants $\epsilon, c_{\mathtt{LB}} \in (0, 1/3)$ and time horizon $T$. The problem instance is defined as follows: budget $B = c_{\mathtt{LB}} T$, mean rewards $r(A_1) = \frac{1 - c_{\mathtt{LB}}}{2}$ and $r(A_2) = 1 - c_{\mathtt{LB}} - \epsilon$, mean consumptions $c(A_1) = c_{\mathtt{LB}} - \epsilon$ and $c(A_2) = 2c_{\mathtt{LB}}$. Parts (1-4) of Assumption 3.8 hold trivially. One can work out that $G_{\mathtt{LAG}} = \epsilon$, so part (4) holds as long as $\epsilon \geq c_{\mathtt{LB}}/\sqrt{T}$.

---

[4]A standard approach for lower-bounding regret in multi-armed bandits is to construct multiple problem instances. A notable exception is the celebrated $\Omega(\log T)$ lower bound in Lai and Robbins [41], which considers one (arbitrary) problem instance, but makes additional assumptions on the algorithm.

**Theorem 3.9.** *Posit an arbitrary time horizon $T$, budget $B$, and $d$ resources (including time). Fix any $3 \times d$ problem instance $\mathcal{I}_0$ which satisfies Assumption 3.8. In part (a), assume that $d = 2$ and $\mathcal{I}_0$ is far from being best-arm-optimal, in the sense that*

$$\text{There exists an optimal solution } \boldsymbol{X}^* \text{ such that } X(A_1) > 2c_{\text{LB}}^4/\sqrt{T} \text{ and } X(A_2) \geq c_{\text{LB}}. \quad (3.15)$$

*In part (b), assume that $d > 2$. For both parts, there exist problem instances $\mathcal{I}, \mathcal{I}'$, which are $\mathcal{O}\left(1/\sqrt{T}\right)$-perturbations of $\mathcal{I}_0$, such that*

$$\text{Any algorithm incurs regret } \texttt{OPT}_{\texttt{FD}} - \mathbb{E}[\texttt{REW}] \geq \Omega(\,c_{\text{LB}}^4 \,\sqrt{T}\,) \text{ on } \mathcal{I} \text{ or } \mathcal{I}' \quad (3.16)$$

For part (a), instance $\mathcal{I}$ has the same expected outcomes as $\mathcal{I}_0$ (but possibly different outcome distributions); we call such problem instances *mean-twins*. For part (b), one can take $\mathcal{I}_0$ to be best-arm-optimal. For both parts, the problem instances $\mathcal{I}, \mathcal{I}'$ require randomized resource consumption.

Both parts follow from a more generic lower bound which focuses on linear independence of per-resource consumption vectors $\boldsymbol{c}_j := (\,c_j(A_1),\, c_j(A_2),\, c_j(\texttt{null})\,) \in [0,1]^3$, resources $j \in [d]$.

**Theorem 3.10.** *Posit an arbitrary time horizon $T$, budget $B$, and $d \geq 2$ resources (including time). Fix any $3 \times d$ problem instance $\mathcal{I}_0$ that satisfies Assumption 3.8 and Eq. (3.15). Assume that the consumption vectors $\boldsymbol{c}_j$, $j \in [d]$ are linearly independent. Then there are instances $\mathcal{I}, \mathcal{I}'$ which are $\epsilon$-perturbations of $\mathcal{I}_0$, with $\epsilon = 2\,c_{\text{LB}}^2/\sqrt{T}$, which satisfy (3.16). In fact, $\mathcal{I}$ is a mean-twin of $\mathcal{I}_0$.*

**Proof Sketch** (see Appendix E for full proof). Let $r(a)$ and $\boldsymbol{c}(a) \in [0,1]^d$ be, resp., the mean reward and the mean resource consumption vector for each arm $a$ for instance $\mathcal{I}_0$. Let $\epsilon = c_{\text{LB}}/\sqrt{T}$.

Problem instances $\mathcal{I}, \mathcal{I}'$ are constructed as follows. For both instances, the rewards of each non-null arm $a \in \{A_1, A_2\}$ are deterministic and equal to $r(a)$. Resource consumption vector for arm $A_1$ is deterministic and equals $\boldsymbol{c}(A_1)$. Resource consumption vector of arm $A_2$ in each round $t$, denoted $\boldsymbol{c}_{(t)}(A_2)$, is a carefully constructed random vector whose expectation is $c(A_2)$ for instance $\mathcal{I}$, and slightly less for instance $\mathcal{I}'$. Specifically, $\boldsymbol{c}_{(t)}(A_2) = \boldsymbol{c}(A_2) \cdot W_t/(1 - c_{\text{LB}})$, where $W_t$ is an independent Bernoulli random variable which correlates the consumption of all resources. We posit $\mathbb{E}[W_t] = 1 - c_{\text{LB}}$ for instance $\mathcal{I}$, and $\mathbb{E}[W_t] = 1 - c_{\text{LB}} - \epsilon$ for instance $\mathcal{I}'$.

Because of the small differences between $\mathcal{I}, \mathcal{I}'$, any algorithm will choose a sufficiently "wrong" distribution over arms sufficiently often. The assumption in Eq. (3.15) and the linear independence condition are needed to ensure that "wrong" algorithm's choices result in large regret. ∎

The corollaries are obtained as follows. For Theorem 3.9(a), problem instance $\mathcal{I}_0$ trivially satisfies all preconditions in Theorem 3.10. Indeed, letting time be resource 1, the per-resource vectors are $\boldsymbol{c}_1 = (0,0,1)$ and $\boldsymbol{c}_2 = (\,\cdot\,,\,\cdot\,,\,0)$, hence they are linearly independent. For Theorem 3.9(b), we use some tricks from the literature to transform the original problem instance $\mathcal{I}_0$ to another instance $\widetilde{\mathcal{I}}_0$ which satisfies Eq. (3.15) and the linear independence condition. The full proof is in Section F.

## 4 Simple regret of `UcbBwK` algorithm

We define *simple regret* in a given round $t$ as $\texttt{OPT}_{\texttt{DP}}/T - r(\boldsymbol{X}_t)$, where $\boldsymbol{X}_t$ is the distribution over arms chosen by the algorithm. The benchmark $\texttt{OPT}_{\texttt{DP}}/T$ generalizes the best-arm benchmark from stochastic bandits. If each round corresponds to a user and the reward is this user's utility, then $\texttt{OPT}_{\texttt{DP}}/T$ is the "fair share" of the total reward. We prove that with `UcbBwK`, all but a few users receive close to their fair share. This holds if $B > \Omega(T) \gg K$, without any other assumptions.

**Theorem 4.1.** *Consider `UcbBwK`. Assume $B \geq \Omega(T)$ and $\eta_{\text{LP}} \leq \frac{1}{2}$. With probability $\geq 1 - O(T^{-3})$, for each $\epsilon > 0$, there are at most $N_\epsilon = \mathcal{O}\left(\frac{K}{\epsilon^2} \log KTd\right)$ rounds $t$ such that $\texttt{OPT}_{\texttt{DP}}/T - r(\boldsymbol{X}_t) \geq \epsilon$.*

To prove Theorem 4.1, we consider another generalization of the "reward-gap", which measures the difference in LP-value compared to $\texttt{OPT}_{\texttt{LP}}$. For distribution $\boldsymbol{X}$ over arms, the *LP-gap* of $\boldsymbol{X}$ is

$$G_{\text{LP}}(\boldsymbol{X}) := \texttt{OPT}_{\texttt{LP}} - V(\boldsymbol{X}), \text{ where } V(\boldsymbol{X}) := (B/T) \cdot r(\boldsymbol{X})/\left(\max_{j \in [d]} c_j(\boldsymbol{X})\right). \quad (4.1)$$

Here, $V(\boldsymbol{X})$ is the value of $\boldsymbol{X}$ in the LP (2.2) after rescaling, so that $\texttt{OPT}_{\texttt{LP}} = \sup_{\boldsymbol{X}} V(\boldsymbol{X})$. Note that $\boldsymbol{X}$ does not need to be feasible for (2.2). It suffices to study the LP-gap because $r(\boldsymbol{X}_t) \geq$

$V(\boldsymbol{X}_t)(1 - \eta_{\text{LP}})$ for each round $t$ with high probability. This holds under the "clean event" in (2.7), because $\boldsymbol{X}_t$ being the solution to the optimistic LP implies $\max_j c_j(\boldsymbol{X}_t) \geq {}^B/_T (1 - \eta_{\text{LP}})$.

Thus, we upper-bound the number of rounds $t$ in which $G_{\text{LP}}(\boldsymbol{X}_t)$ is large. We do this in two steps, focusing on the confidence radius $\text{Rad}_t(\boldsymbol{X}_t)$ as defined in (2.8). First, we upper-bound the number of rounds $t$ with large $\text{Rad}_t(\boldsymbol{X}_t)$. A crucial argument concerns *confidence sums*:

$$\sum_{t \in S} \text{Rad}_t(a_t) \quad \text{and} \quad \sum_{t \in S} \text{Rad}_t(\boldsymbol{X}_t), \tag{4.2}$$

the sums of confidence radii over a given subset of rounds $S \subset [T]$, for, resp., actions $a_t$ and distributions $\boldsymbol{X}_t$ chosen by the algorithm. Second, we upper-bound $G_{\text{LP}}(\boldsymbol{X}_t)$ in terms of $\text{Rad}_t(\boldsymbol{X}_t)$. The details are spelled out in Appendix G.

# 5  Reduction from BwK to stochastic bandits

We improve all regret bounds for UcbBwK algorithm, from worst-case regret to logarithmic regret to simple regret, when the problem instance has some helpful structure. In fact, we provide a general *reduction* which translates insights from stochastic bandits into results on BwK. This reduction works as follows: if prior work on a particular scenario in stochastic bandits provides an improved upper bound on the confidence sums (4.2), this improvement propagates throughout the analyses of UcbBwK. Specifically, suppose $\sum_{t \in S} \text{Rad}_t(a_t) \leq \sqrt{\beta |S|}$ for all algorithms, all subsets of rounds $S \subset [T]$, and some instance-dependent parameter $\beta \ll K$, then UcbBwK satisfies

(i)  worst-case regret $\text{OPT}_{\text{DP}} - \mathbb{E}[\text{REW}] \leq O(\sqrt{\beta T})(1 + \text{OPT}_{\text{DP}}/B)$.
(ii)  Theorem 3.2 holds with $\Psi = \beta\, G_{\text{LAG}}^{-2}$ and regret $\mathcal{O}\left(\beta\, G_{\text{LAG}}^{-1}\right)$ in part (ii).
(iii)  Theorem 4.1 holds with $N_\epsilon = \mathcal{O}\left(\beta\, \epsilon^{-2}\right)$.

Conceptually, this works because confidence sum arguments depend only on the confidence radii, rather than the algorithm that chooses arms, and are about stochastic bandits rather than BwK. The analyses of UcbBwK in [3] and the previous sections use $\beta = K$, the number of arms. The confidence sum bound with $\beta = K$ and results (i, ii, iii) for stochastic bandits follow from the analysis in [10].

We apply this reduction to three well-studied scenarios in stochastic bandits: combinatorial semi-bandits [*e.g.,* 25, 40, 39], linear contextual bandits [*e.g.,* 9, 29, 43, 27, 2], and multinomial-logit (MNL) bandits [*e.g.,* 7, 48, 51, 24]. The confidence-sum bounds are implicit in prior work on stochastic bandits, and we immediately obtain the corresponding extensions for BwK. To put this in perspective, each scenario has lead to a separate paper on BwK [resp., 49, 5, 26], for the worst-case regret bounds alone. We essentially match the worst-case regret bounds from prior work, and obtain new bounds on logarithmic regret and simple regret.[5]  The details are spelled out in Appendix H.

Another reduction from BwK to bandits, found in [35], is very different from ours. It requires a much stronger premise (a regret bound against an adaptive adversary), and only yields worst-case regret bounds. Moreover, it reuses a bandit algorithm as a subroutine, whereas ours reuses a lemma.

# 6  Discussion: significance and novelty

Characterizing (poly-)logarithmic regret rates is a very natural question, and we give a complete answer. The answer consists of positive and negative parts: the positive part requires substantial assumptions, and these assumptions are necessary. The positive result comes "for free" despite the assumptions: it is achieved via UcbBwK and without sacrificing the worst-case performance.

The $O(\log T)$ regret result is well-motivated on its own, even though it requires $d = 2$ and best-arm-optimality and a reasonably small $K = $ #arms. Indeed, problems with $d = 2$ and small $K$ arise in many motivating applications of BwK (see Appendix A), and capture the three challenges of BwK discussed in the Introduction. Moreover, best-arm-optimality is a typical, non-degenerate case. [6]

---

[5]However, we do not provide a generic computationally efficient implementation.

[6]To make this point formal, we focus on $d = 2$ and observe that best-arm-optimality arises with probability at least $p$, for some absolute constant $p > 0$, if expected rewards and expected resource consumptions are drawn independently and uniformly at random. This is a generic fact about LPs, which follows, *e.g.,* from the definition of primal degeneracy in Section 2 of [46], combined with Proposition 2.7.2 in [55].

For lower bounds in terms of Lagrangian gap $G_{\texttt{LAG}}$, we rely on the $\Omega(1/G \cdot \log T)$ regret bound for bandits [41], where $G$ is the reward-gap (since $G_{\texttt{LAG}}$ generalizes reward-gap). In particular, $1/G_{\texttt{LAG}}$ scaling is optimal. No other instance-dependent lower bounds are known for $\texttt{BwK}$. However, Theorem 3.9 implies $\Omega(\sqrt{T})$ regret for some "proper" instances of $\texttt{BwK}$ (*i.e.,* ones with resource consumption) that have small $G_{\texttt{LAG}}$.

Simple regret is a standard performance measure in stochastic bandits, previously not studied for $\texttt{BwK}$. While our result requires $B > \Omega(T) \gg K$, this is the main "parameter regime" of interest in most/all prior work on $\texttt{BwK}$, and a necessity in an important subset of this work [19, 20, 59, 35]. In contrast with stochastic bandits, Theorem 4.1 does not imply logarithmic regret, as per our lower bounds.

The "reduction" result is conceptual rather than technical. We make the point that regret bounds for many extensions of $\texttt{BwK}$ can be derived seamlessly, and identify a mathematical structure which drives these extensions (namely, a bound on confidence sums). In a way, we formalize the intuition that analyses of "optimism under uncertainty" are likely to carry over from stochastic bandits to $\texttt{BwK}$.

We introduce several new concepts and techniques: *Lagrangian gap* (3.1) for logarithmic regret, *LP-gap* (E.2) for analyzing simple regret, and the abstraction of *confidence sums* (4.2). Also, LP-sensitivity arguments appear new in bandit analyses. Both new notions of "gap" satisfy the natural desiderata: they generalize reward-gap, separate the dependence on the problem instance from that on the time horizon $T$ (formally: do not depend on $T$, fixing the $B/T$ ratio), and are "productive", leading to improved results. However, neither notion captures *all* $\texttt{BwK}$ instances with low regret.[7]

---

[7]This should not be surprising per se, as reward-gap does not capture all "nice" bandit instances either. *E.g.,* problem instances with small reward-gap admit $O(\log T)$ regret if they have a likewise small best reward.

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
