# Bandits with Knapsacks beyond the Worst Case (Supplementary Materials)

## Contents

# A  Motivating examples with $d = 2$ and small number of arms

We provide direct motivation for Theorem 3.2, our positive result for $O(\log T)$ regret. Recall that Theorem 3.2 only holds with $d = 2$ resources, and is only meaningful with a reasonably small number of arms $K$ (because the regret bounds are linear in $K$). Such problems arise in many motivating applications of BwK, *e.g.,* as listed in [14, 16]. Below we spell out several stylized examples.

In *dynamic assortment* [50, 7, 26], an algorithm is a seller which chooses among possible assortments of products. In each round, a customer arrives, the algorithm chooses an assortment, and offers this assortment for sale at an exogenously fixed price. If a sale happens, the algorithm receives revenue and consumes some amount of inventory. The following version features $d = 2$ and non-huge $K$: there are $K$ possible offerings for sale, and a limited amount of "raw material" used to manufacture them. Each offering, if sold, consumes some pre-fixed amount of this raw material.[8]

The "inverted" dynamic assortment problem takes the procurement perspective. An algorithm is a budget-limited contractor which chooses among $K$ possible types of offers, *e.g.,* different items to procure from vendors, or different tasks to complete in an online labor market. In each round, a new agent arrives, the algorithm chooses an offer and presents it to the customer at an exogenously fixed price. If the offer is accepted, the contractor receives some utility (*i.e.,* reward) and spends the corresponding amount of money.

In *dynamic pricing* [19, 12, 20, 59] an algorithm is a seller with limited supply of some product, and chooses a price in each round. If this price is accepted, a sale happens, and algorithm receives revenue and spends inventory. Of our interest is the case when the set of possible prices is small and exogenously fixed, *e.g.,* there are a few possible discount levels. Likewise, in *dynamic procurement* [13, 52, 14], an algorithm is a budget-limited contractor who continuously procures some product or service. The algorithm chooses a price in each round. If this price is accepted, a transaction happens, so that the algorithm receives an "item" (*i.e.,* reward of 1) and spends the corresponding amount of money. We focus on the case when there are only a few possible prices, *e.g.,* exogenously fixed levels of premium or surcharge.

Our last example concerns fault-tolerance in systems. Consider a system, either physical or computational, which experiments with different possible policies to process incoming requests. In each time step, it chooses one of the possible policies, and observes the outcome (and there are no lingering effects, *e.g.,* no persistent "system state" that changes over time). The outcome consists of utility for performance-as-usual (*i.e.,* reward), and penalty for various mistakes or faults. Fault-tolerance requirement is expressed as a a "budget" on the total penalty accrued by the algorithm.

# B  Confidence bounds in UcbBwK

Let us fill in the exact specification of the confidence bounds in the UcbBwK algorithm. (This is for the sake of completeness only; as pointed out in Preliminaries, these details do not affect our analysis.)

**Confidence radius.** Given an unknown quantity $\mu$ and its estimator $\widehat{\mu}$, a *confidence radius* is an observable high-confidence upper bound on $|\mu - \widehat{\mu}|$. More formally, it is some quantity $\mathrm{Rad} \in \mathbb{R}_{\geq 0}$ such that it is computable from the algorithm's observations, and $|\mu - \widehat{\mu}| \leq \mathrm{Rad}$ with probability (say) at least $1 - 1/T^3$. Throughout, the estimator $\widehat{\mu}$ is a sample average over all available observations pertaining to $\mu$, unless specified otherwise.

Following the prior work on BwK [12, 16, 3], we use the confidence radius from [38]:

$$f_{\mathtt{rad}}(\widehat{\mu}, N) := \min\left( 1, \ \sqrt{\tfrac{C_{\mathtt{rad}}\,\widehat{\mu}}{\max(1,N)}} + \tfrac{C_{\mathtt{rad}}}{\max(1,N)} \right), \text{ where } C_{\mathtt{rad}} = 3 \cdot \log(KdT), \qquad \text{(B.1)}$$

and $N$ is the number of samples. If $\widehat{\mu}$ is a sample average of $N$ independent random variables with support in $[0, 1]$, and $\mu = \mathbb{E}[\mu]$, then with probability at least $1 - (Kdt)^{-2}$ we have

$$|\widehat{\mu} - \mu| \leq f_{\mathtt{rad}}(\widehat{\mu}, N) \leq 3\, f_{\mathtt{rad}}(\mu, N). \qquad \text{(B.2)}$$

For each arm, we use this confidence radius separately for expected reward of this arm, and expected consumption of each resource.x

---

[8]This framing with raw material(s) — BwK formulations of revenue management problems in which products being sold are separate from raw material(s) being consumed — traces back to Besbes and Zeevi [20].

**Confidence bounds.** Fix arm $a \neq \texttt{null}$, round $t$, and resource $j \neq \texttt{time}$.

Let $S_t(a) = \{s < t : a_s = a\}$ be the set of all previous rounds in which this arm has been chosen, and let $N_t(a) = |S_t(a)|$. Let

$$\hat{r}_t(a) := \tfrac{1}{t} \sum_{s \in S_t(a)} r_s(a) \quad \text{and} \quad \hat{c}_{j,t}(a) := \tfrac{1}{t} \sum_{s \in S_t(a)} c_{j,s}(a) \tag{B.3}$$

denote, resp., the sample average of reward and resource-$j$ consumption of this arm so far.

Define the confidence radii $\mathrm{Rad}_{0,t}(a)$ and $\mathrm{Rad}_{j,t}(a)$ for, resp., expected reward $r(a)$ and resource consumption $c_j(a)$, and the associated upper/lower confidence bounds:

$$\begin{aligned}
r_t^{\pm}(a) &= \mathrm{proj}\left(\,\hat{r}_t(a) \pm \mathrm{Rad}_{0,t}(a)\,\right), & \mathrm{Rad}_{0,t}(a) &:= f_{\texttt{rad}}(\hat{r}_t(a), N_t(a)), \\
c_{j,t}^{\pm}(a) &= \mathrm{proj}\left(\,\hat{c}_{j,t}(a) \pm \mathrm{Rad}_{j,t}(a)\,\right), & \mathrm{Rad}_{j,t}(a) &:= f_{\texttt{rad}}(\hat{c}_{j,t}(a), N_t(a)),
\end{aligned} \tag{B.4}$$

where $\mathrm{proj}(x) := \arg\min_{y \in [0,1]} |y - x|$ denotes the projection into $[0,1]$. Then, the event

$$r(a) \in [r_t^-(a),\ r_t^+(a)] \text{ and } c_j(a) \in [c_{j,t}^-(a),\ c_{j,t}^+(a)], \quad \forall a \in [K], j \in [d-1]. \tag{B.5}$$

holds for each round $t$ with probability (say) at least $1 - \frac{\log(KdT)}{T^4}$ [12].

Note that all confidence radii in (B.4) are upper-bounded by

$$\mathrm{Rad}_t(a) := f_{\texttt{rad}}(1, N_t(a)), \tag{B.6}$$

which is a version of a more standard confidence radius $\widetilde{O}(1/\sqrt{N_t(a)})$.

There is no uncertainty on the time resource and the null arm. So, we set $\mathrm{Rad}_{\texttt{time}, t}(\cdot) = 0$ and $c_{\texttt{time}, t}^{\pm}(\cdot) = B/T$, and $\mathrm{Rad}_{0,t}(\texttt{null}) = \mathrm{Rad}_{j,t}(\texttt{null}) = r^{\pm}(\texttt{null}) = c_{j,t}^{\pm}(\texttt{null}) = 0$.

# C    LP Sensitivity: proof of Lemma 3.3

We focus on the sensitivity *of the support of the optimal solution*. We build on some well-known results, which we state below in a convenient form (and provide a proof for completeness). We use the textbook material from Bertsimas and Tsitsiklis [18].

Throughout this appendix, we consider a best-arm-optimal problem instance with best arm $a^*$. Let $\boldsymbol{X}^*$ denote the optimal solution for the linear program (2.2). Recall that the support of $\boldsymbol{X}^*$ is either $\{a^*\}$ or $\{a^*, \texttt{null}\}$. We consider perturbations in the *rescaled LP*:

$$\begin{aligned}
\text{maximize} \qquad & \boldsymbol{X} \cdot \boldsymbol{r} & \text{such that} \\
& \boldsymbol{X} \in [0,1]^K \\
& \boldsymbol{X} \cdot \boldsymbol{1} = 1 \\
\forall j \in [d-1] \qquad & \boldsymbol{X} \cdot \boldsymbol{c}_j \leq (B/T)(1 - \eta_{\texttt{LP}}) \\
& \boldsymbol{X} \cdot \boldsymbol{c}_d \leq B/T.
\end{aligned} \tag{C.1}$$

Recall that $\boldsymbol{r}, \boldsymbol{c}_j \in [0,1]^K$ are vectors of expected rewards and expected consumption of resource $j$. The $d$-th resource is time. The rescaling parameter $\eta_{\texttt{LP}}$ is given in Eq. (2.5).

Let $\texttt{OPT}_{\texttt{LP}}^{\texttt{sc}}$ denote the value of this LP; it is easy to see that $\texttt{OPT}_{\texttt{LP}}^{\texttt{sc}} = (1 - \eta_{\texttt{LP}})\,\texttt{OPT}_{\texttt{LP}}$.

We observe that $a^*$ is the best arm for the rescaled LP, too, because $G_{\texttt{LAG}}$ is large enough. Call a distribution over arms *null-degenerate* if its support includes exactly one non-null arm.

**Claim C.1.** *The rescaled LP* (C.1) *has a null-degenerate optimal solution with non-null arm* $a^*$.

*Proof.* From the theory in [18, Ch.5], if the optimal basis to LP (2.2) remains *feasible* to the rescaled LP (C.1) then the basis is also optimal to this LP. This is because LP (C.1) is obtained by a small perturbation to the right-hand side values in LP (2.2). Let $\boldsymbol{X}^*$ denote the optimal solution to LP (2.2). From assumption this is a null-degenerate optimal solution. Using the same analysis in [18, Ch. 4.4] we only have to show that the perturbation is smaller than $X^*(a^*)$. Since the perturbation is $\frac{B\eta_{\texttt{LP}}}{T} \leq \frac{3\sqrt{B}\log(KTd)}{T}$ while $X^*(a^*) > \frac{3\sqrt{B}\log(KTd)}{T}$, this perturbation does not change the basis. Thus, the rescaled LP has a null-degenerate optimal solution. $\square$

**Claim C.2.** *Let $\boldsymbol{\lambda}^*$ denote the vector of the optimal dual solution to the LP* (2.2). *Then*

$$G_{LAG}(a) = \tfrac{T}{B} \sum_{j \in [d]} \lambda_j^* c_j(a) - r(a). \tag{C.2}$$

*Proof.* From Eq. (3.1) we have the following.

$$
\begin{aligned}
G_{\texttt{LAG}}(a) &:= \mathcal{L}(\boldsymbol{X}^*, \boldsymbol{\lambda}^*) - \mathcal{L}(\boldsymbol{X}_a, \boldsymbol{\lambda}^*) \\
&= \boldsymbol{r}(\boldsymbol{X}^*) - \tfrac{T}{B} \sum_{j \in [d]} \lambda_j^* \, \boldsymbol{c}_j(\boldsymbol{X}^*) + \tfrac{T}{B} \sum_{j \in [d]} \lambda_j^* c_j(a) - r(a).
\end{aligned}
$$

Consider the dual of the LP (2.2). It can be seen that the objective of this dual is $\sum_{j \in [d]} \lambda_j$. It follows that $\texttt{OPT}_{\texttt{LP}} = \sum_{j \in [d]} \lambda_j^*$ by strong duality [22, Section 5.2.3]. As proved in [35], $\mathcal{L}(\boldsymbol{X}^*, \boldsymbol{\lambda}^*) = \texttt{OPT}_{\texttt{LP}}$. Thus,

$$\textstyle\sum_{j \in [d]} \lambda_j^* = \texttt{OPT}_{\texttt{LP}} = \mathcal{L}(\boldsymbol{X}^*, \boldsymbol{\lambda}^*) = \boldsymbol{r}(\boldsymbol{X}^*) - \tfrac{T}{B} \sum_{j \in [d]} \lambda_j^* \, \boldsymbol{c}_j(\boldsymbol{X}^*) + \sum_{j \in [d]} \lambda_j^*.$$

Therefore, $\boldsymbol{r}(\boldsymbol{X}^*) = \tfrac{T}{B} \sum_{j \in [d]} \lambda_j^* \, \boldsymbol{c}_j(\boldsymbol{X}^*)$, which implies (C.2). $\qquad\square$

Claim 3.3 easily follows from the following standard result by letting $\delta(a) = \mathrm{Rad}_t(a)$.

**Theorem C.3** (perturbation). *Posit only one resource other than time (i.e., $d = 2$). Consider a perturbation of the rescaled LP (C.1), where the reward vector $\boldsymbol{r}$ is replaced with $\tilde{\boldsymbol{r}}$, and the consumption vector $\boldsymbol{c}_1$ for the non-time resource is replaced with $\tilde{\boldsymbol{c}}_1$. Let $\tilde{\boldsymbol{X}}^*$ be its optimal solution. Assume $0 \leq \tilde{\boldsymbol{r}} - \boldsymbol{r} \leq \boldsymbol{\delta}$ and $0 \leq \boldsymbol{c}_1 - \tilde{\boldsymbol{c}}_1 \leq \boldsymbol{\delta}$, for some vector $\boldsymbol{\delta} \in [0,1]^K$. Then for each arm $a \neq a^*$,*

$$\delta(a) > G_{LAG}(a) \quad \text{if} \quad a \in \mathrm{supp}(\tilde{\boldsymbol{X}}^*).$$

*Proof.* Let $\lambda_1^* \geq 0$ denote the dual variable corresponding to the single resource. Note that since $\texttt{OPT}_{\texttt{LP}} \leq 1$ and the dual vector $\boldsymbol{\lambda}^* \geq \boldsymbol{0}$ coordinate wise, we have $\lambda_1^* \leq 1$. From [18, Ch. 5.1] on local sensitivity when non-basic column of $A$ is changed, we have that the maximum allowable change to any single column $\delta(a) \leq \frac{\tilde{c}(a)}{\lambda_1^*}$ where $\tilde{c}(a)$ is the reduced-cost for the simplex algorithm, as defined in [18]. We will show that $\tilde{c}(a) = G_{\texttt{LAG}}(a)$. Thus, if $\delta(a) \leq \frac{\tilde{c}(a)}{\lambda_1^*} = \frac{G_{\texttt{LAG}}(a)}{\lambda_1^*}$ we have that the basis remains unchanged. Likewise from Bertsimas and Tsitsiklis [18, Ch. 5], the maximum allowed perturbation $\delta(a)$ on the reward $r(a)$ for the basis to remain unchanged is $\delta(a) \leq \tilde{c}(a)$. Combining these two we get the "*if*" part of the theorem.

It remains to prove that the reduced cost $\tilde{c}(a) = G_{\texttt{LAG}}(a)$. After converting the linear program to the standard form as required in [18], the reduced-cost $\tilde{c}(a)$ is given by the expression $\frac{T}{B(1-\eta_{\texttt{LP}})} \sum_{j \in [d]} c_j(a) \tilde{\lambda}_j^* - r(a)$ where $\tilde{\boldsymbol{\lambda}}^*$ is the optimal dual solution to LP (C.1). Note that $\boldsymbol{\lambda}^* := \left(\frac{1}{1-\eta_{\texttt{LP}}}\right) \tilde{\boldsymbol{\lambda}}^*$ is an optimal solution to the dual of the LP (2.2). Thus, plugging it into the definition of reduced cost and combining it with Claim C.2 we have that

$$\tilde{c}(a) = \frac{T}{B} \sum_{j \in [d]} \lambda_j^* c_j(a) - r(a) = G_{\texttt{LAG}}(a). \qquad\square$$

# D Various technicalities from Sections 3

## D.1 Standard tools

We rely on some standard tools, which we state below for the sake of convenience.

**Theorem D.1** (Wald's identity). *Let $X_i : \ i \in \mathbb{N}$ be i.i.d. real-valued random variables, adapted to filtration $\mathcal{F}_i : \ i \in \mathbb{N}$. Let $N$ be a stopping time relative to the same filtration. Then*

$$\mathbb{E}[X_1 + X_2 + \ldots + X_N] = \mathbb{E}[X_i] \cdot \mathbb{E}[N].$$

**Theorem D.2** (Optimal Stopping Theorem). *Let $X_i : \ i \in \mathbb{N}$ be a martingale sequence with $\mathbb{E}[X_0] = 0$ adapted to filtration $\mathcal{F}_i : \ i \in \mathbb{N}$. Let $N$ be a stopping time relative to the same filtration. Then we have that $\mathbb{E}[X_N] = 0$.*

**Theorem D.3** ([37, 12]). *Let $Z_1, Z_2, , \ldots , Z_T$ be a martingale w.r.t. filtration $(\mathcal{F}_t)_{t \in [T]}$, such that $|Z_t| \leq c$ for all $t \in [T]$. Let $\mu := \frac{1}{T} \sum_{t \in [T]} \mathbb{E}[Z_t \mid \mathcal{F}_{t-1}]$. Then,*

$$\Pr \left[ \left| \sum_{t \in [T]} Z_t - \mu T \right| > \sqrt{2\mu T c^2 \ln \tfrac{T}{\delta}} \right] \leq \delta.$$

## D.2 Proof of Eq. (3.6)

Let $\tau$ denote the stopping time of the algorithm that chooses arm $a^*$ in every time-step, given that the total budget is $B_0$, $T_0$ on the two resources. From definition we have $\mathtt{REW}(a^* \mid B_0, T_0) = \sum_{t \in [\tau]} r_t(a^*)$. Using Wald's identity (Theorem D.1), we have that $\mathbb{E}[\mathtt{REW}(a^* \mid B_0, T_0)] = \mathbb{E}[\tau] \ r(a^*)$.

Let $B_0, T_0$ denote the budget remaining for the two resources. By definition, we have that $\tau \geq T_0$ and $\sum_{t \in [\tau]} c_t(a^*) \geq B_0$. Using the Wald's identity (Theorem D.1) we have that $\mathbb{E}[\sum_{t \in [\tau]} c_t(a^*)] = \mathbb{E}[\tau] c(a^*)$. Thus, we have $\mathbb{E}[\tau] \geq \min \left\{ T_0, \frac{B_0}{c(a^*)} \right\} \geq \min \{T_0, B_0\}$. Therefore, we obtain the following.

$$\mathbb{E}[\mathtt{REW}(a^* \mid B_0, T_0)] = \mathbb{E}[\tau] r(a^*) > \left( \frac{\min \{T_0, B_0\}}{\max\{\frac{B}{T}, c(a^*)\}} \right) r(a^*), \quad \text{and} \tag{D.1}$$

$$\mathbb{E}[\mathtt{REW}(a^* \mid B)] = \mathbb{E}[\tau_B] r(a^*) \leq \left( \frac{B}{\max\{\frac{B}{T}, c(a^*)\}} \right) r(a^*). \tag{D.2}$$

Combining Equations (D.1) and (D.2), we get Eq. (3.6).

## D.3 Proof of Eq. (3.9)

We now modify the above proof to get the tighter lower-bound in Eq. (3.9). Let $T_0$, $B_0$ denote the expected remaining time and budget (respectively) and let $\tau$ denote the (random) stopping time of the algorithm that chooses arm $a^*$ in every time-step given $T_0$ time-steps and $B_0$ budget. This implies that we have, $\mathbb{E}[\sum_{t \in [\tau]} c_t(a^*)] \geq B_0$ and $\mathbb{E}[\tau] \geq T_0$. From Theorem D.1, this implies that we have $\mathbb{E}[\tau] c(a^*) \geq B_0$ and $\mathbb{E}[\tau] \geq T_0$. This implies that $\mathbb{E}[\tau] \geq \min\{T_0, \frac{B_0}{c(a^*)}\}$.

Similar to Eq. (D.1) and Eq. (D.2) we obtain the following.

$$\mathbb{E}[\mathtt{REW}(a^* \mid B_0, T_0)] = \mathbb{E}[\tau] r(a^*) > \min\{T_0, \tfrac{B_0}{c(a^*)}\} r(a^*), \quad \text{and} \tag{D.3}$$

$$\mathbb{E}[\mathtt{REW}(a^* \mid B_0 = B, T_0 = T)] = \mathtt{OPT_{FD}} \leq \left( \frac{B}{\max\{\frac{B}{T}, c(a^*)\}} \right) r(a^*). \tag{D.4}$$

Combining Equations (D.3) and (D.4), we get Eq. (3.9).

## D.4 Lower bound on Lagrange gap: Proof of Eq. (3.10)

We will use Eq. (3.4) and some standard properties of linear programming.

Assume $c(a^*) < \frac{B}{T}$. Using complementary slackness theorem on LP (2.2), this implies that $\lambda_1^* = 0$. Moreover, note that the objective in the dual of LP (2.2) is $\lambda_0^* + \lambda_1^* = \lambda_0^*$. The optimal value of the primal LP (2.2) is $r(a^*)$ since, $X(a^*) = 1$ is the optimal solution to the LP. This implies that $\lambda_0^* = r(a^*) \geq \frac{\text{OPT}_{\text{FD}}}{T}$. Substituting this into Eq. (3.4) gives the first inequality in Eq. (3.10).

Now assume $c(a^*) > \frac{B}{T}$. Again, as above complementary slackness theorem on LP (2.2), this implies that $\lambda_0^* = 0$. Thus, $G_{\text{LAG}}(a) = \frac{T}{B} \cdot \lambda_1^* \cdot c(a) - r(a)$. Using the dual objective function $\lambda_0^* + \lambda_1^* = \lambda_1^*$ combined with strong duality, this implies that $\lambda_1^* = \frac{\text{OPT}_{\text{LP}}}{T} \geq \frac{\text{OPT}_{\text{FD}}}{T}$. Plugging this back into Eq. (3.4) gives the second inequality in Eq. (3.10).

## D.5 Martingale arguments: Proof of Eq. (3.8)

For the proof of Eq. (3.8), we use the well-known theorem on optimal stopping time of martingales (Theorem D.2). Fix an arm $a \in [K]$. For any subset $S \subseteq [T]$ of rounds let $N_S(a)$, $r_S(a)$ and $c_S(a)$ denote the number of times arm $a$ is chosen, the total realized rewards for arm $a$ and the total realized consumption of arm $a$, respectively. Let $\tau$ denote the (random) stopping time of a BwK algorithm with (random) budget $B$ and time $T$. Then we have the following claim.

**Claim D.4.** *For a random stopping time $\tau$, for every arm $a \in [K]$ we have the following.*

$$\mathbb{E}\left[r_{[\tau]}(a)\right] = r(a) \cdot \mathbb{E}[N_{[\tau]}(a)]. \tag{D.5}$$

$$\mathbb{E}\left[c_{[\tau]}(a)\right] = c(a) \cdot \mathbb{E}[N_{[\tau]}(a)]. \tag{D.6}$$

*Proof.* We will prove the equality in Eq. (D.5); the one in Eq. (D.6) follows. Consider $r_{[\tau]}(a)$. By definition this is equal to $\sum_{t \in [\tau]} r_t(a) \cdot \mathbb{I}[a_t = a]$. Let $A_t := \mathbb{I}[a_t = a]$ denote the random variable corresponding to the event that arm $a$ is chosen at time $t$. Define the random variable

$$Y_t := \sum_{t' \leq t} A_{t'} r_{t'}(a) - \mathbb{E}_{t'}\left[A_{t'} r_{t'}(a)\right],$$

where $\mathbb{E}_t[.]$ denotes the conditional expectation given the random variables $A_1, A_2, \ldots, A_{t-1}$. It is easy to see that the sequences $\{X_t\}_{t \in [\tau]}$, $\{Y_t\}_{t \in [\tau]}$ and $\{Z_t\}_{t \in [\tau]}$ forms a martingale sequence. Thus, we will apply the optimal stopping theorem (Theorem D.2) at time $\tau$, we have the following.

$$\mathbb{E}[Y_\tau] = \mathbb{E}\left[\sum_{t' \leq \tau} A_{t'} r_{t'}(a)\right] - \mathbb{E}\left[\sum_{t' \leq \tau} \mathbb{E}_{t'}\left[A_{t'} r_{t'}(a)\right]\right] = 0. \tag{D.7}$$

Consider the term $\mathbb{E}\left[\sum_{t' \leq \tau} \mathbb{E}_{t'}\left[A_{t'} r_{t'}(a)\right]\right]$ in Eq. (D.7). This can be simplified to $\mathbb{E}\left[\sum_{t' \leq \tau} r(a) \cdot \Pr[a_{t'} = a]\right]$. Consider the following random variable

$$Z_t := \sum_{t' \leq t} \Pr[a_{t'} = a] - \mathbb{E}_{t'}[\Pr[a_{t'} = a]].$$

Note that $\sum_{t' \leq t} \mathbb{E}_{t'}[\Pr[a_{t'} = a]] = N_{[t]}(a)$. Thus, using Theorem D.2 on the sequence $Z_t$ at the stopping time $\tau$, we obtain $\mathbb{E}\left[\sum_{t' \leq \tau} \Pr[a_{t'} = a]\right] = \mathbb{E}[N_{[\tau]}(a)]$.

Thus, the term $\mathbb{E}\left[\sum_{t' \leq \tau} \mathbb{E}_{t'}\left[A_{t'} r_{t'}(a)\right]\right]$ in Eq. (D.7) simplifies to $r(a) \cdot N_{[\tau]}(a)$ which gives the required equality in Eq. (D.5). $\square$

We will now use Claim D.4 to prove Eq. (3.8). Recall that REW$(a \mid B(a), T(a))$ denotes the total contribution to the reward by the BwK algorithm by playing arm $a$ with a (random) resource consumption of $B(a)$ and time steps of $T(a)$. Let $\tau$ be the (random) stopping time of this algorithm.

By definition we have that $N_{[\tau]}(a) = T(a)$. Thus, $\mathbb{E}[N_{[\tau]}(a)] = \mathbb{E}[T(a)]$. From Eq. (D.6), we also have that $\mathbb{E}[N_{[\tau]}(a)] = \frac{\mathbb{E}[c_{[\tau]}(a)]}{c(a)}$. From the definition of $B(a)$ we have, $B(a) = c_{[\tau]}(a)$ and thus, $\mathbb{E}[B(a)] = \mathbb{E}[c_{[\tau]}(a)]$. Thus, this implies that $\mathbb{E}[N_{[\tau]}(a)] = \min\{T(a), \frac{\mathbb{E}[B(a)]}{c(a)}\}$.

Consider $\mathbb{E}[\texttt{REW}(a)] = \mathbb{E}[\texttt{REW}(a \mid B(a), T(a))]$.

$$
\begin{aligned}
\mathbb{E}[\texttt{REW}(a \mid B(a), T(a))] &= \mathbb{E}\left[r_{[\tau]}(a)\right] \\
&= r(a) \cdot \mathbb{E}[N_{[\tau]}(a)] \qquad\qquad \textit{(From Eq. (D.5))} \\
&= r(a) \cdot \min\{T(a), \tfrac{\mathbb{E}[B(a)]}{c(a)}\}
\end{aligned}
\tag{D.8}
$$

Now, consider $\texttt{LP}(a \mid \mathbb{E}[B(a)], \mathbb{E}[T(a)])$. This value is equal to,

$$
\begin{aligned}
\mathbb{E}[\texttt{REW}(a \mid \mathbb{E}[B(a)], \mathbb{E}[T(a)])] &= \frac{r(a)}{\max\{\mathbb{E}[B(a)]/\mathbb{E}[T(a)], c(a)\}} \cdot \frac{\mathbb{E}[B(a)]}{\mathbb{E}[T(a)]} \\
&= r(a) \cdot \min\left\{\mathbb{E}[T(a)], \tfrac{\mathbb{E}[B(a)]}{c(a)}\right\}.
\end{aligned}
$$

Note that the last equality is same as the RHS in Eq. (D.8).

# E  Proof of Theorem 3.10: generic $\sqrt{T}$ lower bound

**Preliminaries.** We rely on a well-known information-theoretic result for multi-armed bandits: essentially, no algorithm can reliably tell apart two bandit instances at time $T$ if they differ by at most $O(1/\sqrt{T})$.[9] We formulate this result in a way that is most convenient for our applications.

**Lemma E.1.** *Consider multi-armed bandits with Bernoulli rewards. Fix $\epsilon > 0$ and two problem instances $\mathcal{I}, \mathcal{I}'$ such that the mean reward of each arm differs by at most $\epsilon$ between $\mathcal{I}$ and $\mathcal{I}'$. Suppose some bandit algorithm outputs distribution $\boldsymbol{Y}_t$ over arms at time $t \leq c/\epsilon^2$, for a sufficiently small absolute constant c. Let H be an arbitrary Lebesgue-measurable set of distributions over arms. Then either $\Pr[\boldsymbol{Y}_t \in H \mid \mathcal{J}_t = \mathcal{I}] > 1/4$ or $\Pr[\boldsymbol{Y}_t \notin H \mid \mathcal{J}_t = \mathcal{I}'] > 1/4$ holds.*

Applying Lemma E.1 to bandits with knapsacks necessitates some subtlety. First, the rewards in the lemma will henceforth be called *quasi-rewards*, as they may actually correspond to consumption of a particular resource. Second, while a BwK algorithm receives multi-dimensional feedback in each round, the feedback other than the quasi-rewards will be the same (in distribution) for both problem instances, and hence can be considered a part of the algorithm. Third, distribution $\boldsymbol{Y}_t$ will be the conditional distribution over arms chosen by the BwK algorithm in round $t$ given the algorithm's observations so far; we will assume this without further mention. Fourth, we will need to specify the set $H$ of distributions (which will depend on a particular application).

Consider the rescaled LP (C.1) with $\eta_{\text{LP}} := 6 * \texttt{OPT}_{\text{LP}} \sqrt{\frac{\log dT}{B}}$; we use this $\eta_{\text{LP}}$ throughout this proof. Let $\texttt{OPT}_{\text{LP}}^{\text{sc}}$ be the value of this LP. We prove the lower bound using $\texttt{OPT}_{\text{LP}}^{\text{sc}}$ as a benchmark. This suffices by the following claim from prior work: [10]

**Claim E.2** (Immorlica et al. [35]). *$OPT_{LP}^{sc} \leq OPT_{FD}$ for $\eta_{\text{LP}} := 6 \cdot OPT_{LP} \sqrt{\frac{\log dT}{B}}$.*

**Problem instances.** Let $\boldsymbol{r}(a)$ and $\boldsymbol{c}(a) \in [0,1]^d$ be, resp., the mean reward and the mean resource consumption vector for each arm $a$ for instance $\mathcal{I}_0$. Let $\epsilon = c_{\text{LB}}/\sqrt{T}$.

Problem instances $\mathcal{I}, \mathcal{I}'$ are constructed as specified in the proof sketch; we repeat it here for the sake of convenience. For both instances, the rewards of each non-null arm $a \in \{A_1, A_2\}$ are deterministic and equal to $r(a)$. Resource consumption vector for arm $A_1$ is deterministic and equals $\boldsymbol{c}(A_1)$. Resource consumption vector of arm $A_2$ in each round $t$, denoted $\boldsymbol{c}_{(t)}(A_2)$, is a carefully

---

[9]This strategy for proving lower bounds in multi-armed bandits goes back to Auer et al. [11]. Lemma E.1 is implicit in Auer et al. [11], see Slivkins [54, Lemma 2.9] for exposition.

[10]Claim E.2 is a special case of Lemma 8.6 in Immorlica et al. [35] for $\tau^* = T$ and the reward/consumption for each arm, each resource and each time-step replaced with the mean reward/consumption.

constructed random vector whose expectation is $c(A_2)$ for instance $\mathcal{I}$, and slightly less for instance $\mathcal{I}'$. Specifically, $\boldsymbol{c}_{(t)}(A_2) = \boldsymbol{c}(A_2) \cdot W_t/(1 - c_{\text{LB}})$, where $W_t$ is an independent Bernoulli random variable which correlates the consumption of all resources. We posit $\mathbb{E}[W_t] = 1 - c_{\text{LB}}$ for instance $\mathcal{I}$, and $\mathbb{E}[W_t] = 1 - c_{\text{LB}} - \epsilon$ for instance $\mathcal{I}'$.

**Main derivation.** From the premise of the theorem (Eq. (3.15)), problem instance $\mathcal{I}$ admits an optimal solution $\boldsymbol{X}^*$ that is substantially supported on both non-null arms. Let $\boldsymbol{X}^*_{\mathcal{I}}, \boldsymbol{X}^*_{\mathcal{I}'}$ denote the optimal solutions to the scaled LP, instantiated for instances $\mathcal{I}, \mathcal{I}'$ respectively.

The proof proceeds as follows. We first prove certain properties of distributions $\boldsymbol{X}^*_{\mathcal{I}}$ and $\boldsymbol{X}^*_{\mathcal{I}'}$. We then use these properties and apply Lemma E.1 with suitable quasi-rewards to complete the proof of the lower-bounds.

Since we modify the mean consumption of all resources for one arm in $\mathcal{I}'$ this implies that $\boldsymbol{X}^*_{\mathcal{I}} \neq \boldsymbol{X}^*_{\mathcal{I}'}$. From assumption 3.8-(3.8) we have that $G_{\text{LAG}} \geq c_{\text{LB}}/\sqrt{T}$. From the premise of the theorem, we have that the mean vector of consumptions for the resources $j \in [d]$ are all linearly independent. Thus, we can apply sensitivity theorem C.3 to conclude that the support of the solution $\boldsymbol{X}^*_{\mathcal{I}'}$ is same as $\boldsymbol{X}^*_{\mathcal{I}}$.

Moreover, from the linear independence of the consumption vectors and Eq. (3.15). combined with standard LP theory (see chapter 4 on duality in [18]) we have that there exists a resource $j^* \in [d]$ such that the optimal solution $\boldsymbol{X}^*_{\mathcal{I}}$ satisfies the resource constraint with equality.

In what follows, we denote the vector $\boldsymbol{c}$ as a shorthand for $\boldsymbol{c}_{j^*}$ (*i.e.,* we drop the index $j^*$). Note that from the perturbation we have that $c(A_1) < c(A_2)$. Thus, for some $\delta > 0$ we have $X^*_{\mathcal{I}'}(A_1) = X^*_{\mathcal{I}}(A_1) - \delta$ and $X^*_{\mathcal{I}'}(A_2) = X^*_{\mathcal{I}}(A_2) + \delta$. Let $\|\boldsymbol{X}\|$ denote the $\ell_1$-norm of a given distribution $\boldsymbol{X}$. Thus, we have

$$\|\boldsymbol{X}^*_{\mathcal{I}} - \boldsymbol{X}^*_{\mathcal{I}'}\| = 2\delta. \tag{E.1}$$

Given any distribution $\boldsymbol{Y}$ over the arms, let

$$V_{\text{sc}}(\boldsymbol{Y}) := (1 - \eta_{\text{LP}}) \cdot {}^{B}/_{T} \cdot r(\boldsymbol{Y})/ \left( \max_{j \in [d]} c_j(\boldsymbol{Y}) \right). \tag{E.2}$$

This is the value of $\boldsymbol{Y}$ in the rescaled LP (C.1), where $\boldsymbol{Y}$ itself is rescaled to make it LP-feasible (and as large as possible). Note that $V_{\text{sc}}(\boldsymbol{Y}) = (1 - \eta_{\text{LP}}) V(\boldsymbol{Y})$, where $V(\boldsymbol{Y})$ is the value of the original LP, as defined in (E.2). Also, $\mathtt{OPT}^{\text{sc}}_{\text{LP}} = \sup_{\boldsymbol{Y}} V_{\text{sc}}(\boldsymbol{Y})$.

By a slight abuse of notation, let $V_{\text{sc}}(\boldsymbol{Y}), V'_{\text{sc}}(\boldsymbol{Y})$ be the value of $V_{\text{sc}}(\boldsymbol{Y})$ corresponding to instances $\mathcal{I}$ and $\mathcal{I}'$ respectively.

We use the following two claims in the proof of our lower-bound. Claim E.3 states that if a distribution is close to the optimal distribution for instance $\mathcal{I}$ then it is also far from the optimal distribution for $\mathcal{I}'$. Claim E.4 states that if a distribution is far from the optimal distribution, then playing from that distribution also incurs large instantaneous regret. Both claims have nothing to do with particular algorithms.

**Claim E.3.** *Fix distribution* $\boldsymbol{Y} \in \Delta^3$ *and* $\epsilon < 1$. *If* $\|\boldsymbol{X}^*_{\mathcal{I}} - \boldsymbol{Y}\| < \epsilon \cdot c^2_{\text{LB}}$ *then* $\|\boldsymbol{X}^*_{\mathcal{I}'} - \boldsymbol{Y}\| \geq \epsilon \cdot c^2_{\text{LB}}$.

**Claim E.4.** *Fix distribution* $\boldsymbol{Y} \in \Delta^3$ *and* $\epsilon < 1$. *If* $\|\boldsymbol{X}^*_{\mathcal{I}} - \boldsymbol{Y}\| \geq \epsilon \cdot c^2_{\text{LB}}$ *then* $V_{sc}(\boldsymbol{X}^*_{\mathcal{I}}) - V_{sc}(\boldsymbol{Y}) \geq \epsilon \cdot \frac{c^3_{\text{LB}}}{2}$. *Likewise, if* $\|\boldsymbol{X}^*_{\mathcal{I}'} - \boldsymbol{Y}\| \geq \epsilon \cdot c^2_{\text{LB}}$ *then* $V'_{sc}(\boldsymbol{X}^*_{\mathcal{I}'}) - V'_{sc}(\boldsymbol{Y}) \geq \epsilon \cdot \frac{c^3_{\text{LB}}}{2}$.

We now invoke Lemma E.1 with the quasi-rewards at each time-step determined by the consumption of the resource $j^*$.

Define the set,

$$\mathcal{H} := \left\{ \boldsymbol{Y} : \|\boldsymbol{X}^*_{\mathcal{I}} - \boldsymbol{Y}\| \geq \epsilon \cdot c^2_{\text{LB}} \right\}, \tag{E.3}$$

to complete the proof Theorem 3.10. Consider an arbitrary algorithm $\mathtt{ALG}$. We consider two cases: $\mathcal{J} = \mathcal{I}$ and $\mathcal{J} = \mathcal{I}'$, which denote the instance that satisfies the conclusion of this lemma for at least $\frac{T}{2}$ rounds for $T := \frac{c_{\text{LB}}}{\epsilon^2}$.

Let $\mathcal{J} = \mathcal{I}$. Let $\mathcal{T}$ denote the set of time-steps $t \in [T]$ such that $\mathcal{J}_t = \mathcal{I}$ and $\boldsymbol{Y}_t \in \mathcal{H}$. Then, the expected regret of ALG can be lower-bounded by,

$$\mathbb{E}\left[\sum_{t \in \mathcal{T}} V_{\mathrm{sc}}(\boldsymbol{X}_{\mathcal{I}}^*) - V_{\mathrm{sc}}(\boldsymbol{Y}_t)\right] = \mathbb{E}\left[\sum_{t \in \mathcal{T}:\, \|\boldsymbol{X}_{\mathcal{I}}^* - \boldsymbol{Y}_t\| \geq \epsilon \cdot c_{\mathrm{LB}}^2} V_{\mathrm{sc}}(\boldsymbol{X}_{\mathcal{I}}^*) - V_{\mathrm{sc}}(\boldsymbol{Y}_t)\right] \qquad \textit{(by Eq. (E.3))}$$

$$\geq \mathbb{E}\left[\sum_{t \in \mathcal{T}} \epsilon \cdot \frac{c_{\mathrm{LB}}^3}{2}\right] \qquad \textit{(by Eq. (E.4))}$$

$$\geq T/4 \cdot \epsilon \cdot \frac{c_{\mathrm{LB}}^3}{2} \qquad \textit{(by Lemma E.1)}$$

$$\geq O\left(c_{\mathrm{LB}}^4 \cdot \sqrt{T}\right). \qquad \textit{(Since } \epsilon = \frac{c_{\mathrm{LB}}}{\sqrt{T}}\textit{)}$$

We use a similar argument when $\mathcal{J} = \mathcal{I}'$. Let $\mathcal{T}'$ denote the set of time-steps $t \in [T]$ such that $\mathcal{J}_t = \mathcal{I}'$ and $\|\boldsymbol{X}_{\mathcal{I}'}^* - \boldsymbol{Y}_t\| \geq \epsilon \cdot c_{\mathrm{LB}}^2$. The expected regret of ALG can be lower-bounded by,

$$\mathbb{E}\left[\sum_{t \in \mathcal{T}'} V_{\mathrm{sc}}'(\boldsymbol{X}_{\mathcal{I}'}^*) - V_{\mathrm{sc}}'(\boldsymbol{Y}_t)\right] = \mathbb{E}\left[\sum_{t \in \mathcal{T}':\, \|\boldsymbol{X}_{\mathcal{I}'}^* - \boldsymbol{Y}_t\| \geq \epsilon \cdot c_{\mathrm{LB}}^2} V_{\mathrm{sc}}'(\boldsymbol{X}_{\mathcal{I}'}^*) - V_{\mathrm{sc}}'(\boldsymbol{Y}_t)\right]$$

$$\geq \mathbb{E}\left[\sum_{t \in \mathcal{T}':\, \|\boldsymbol{X}_{\mathcal{I}}^* - \boldsymbol{Y}_t\| < \epsilon \cdot c_{\mathrm{LB}}^2} V_{\mathrm{sc}}'(\boldsymbol{X}_{\mathcal{I}'}^*) - V_{\mathrm{sc}}'(\boldsymbol{Y}_t)\right] \qquad \textit{(by Claim E.3)}$$

$$= \mathbb{E}\left[\sum_{t \in \mathcal{T}':\, \boldsymbol{Y}_t \notin \mathcal{H}} V_{\mathrm{sc}}'(\boldsymbol{X}_{\mathcal{I}'}^*) - V_{\mathrm{sc}}'(\boldsymbol{Y}_t)\right] \qquad \textit{(by Eq. (E.3))}$$

$$\geq \mathbb{E}\left[\sum_{t \in [T]:\, \boldsymbol{Y}_t \notin \mathcal{H}} \epsilon \cdot \frac{c_{\mathrm{LB}}^3}{2}\right] \qquad \textit{(by Eq. (E.4))}$$

$$\geq T/4 \cdot \epsilon \cdot \frac{c_{\mathrm{LB}}^3}{2} \qquad \textit{(by Lemma E.1)}$$

$$\geq O\left(c_{\mathrm{LB}}^4 \cdot \sqrt{T}\right). \qquad \textit{(Since } \epsilon = \frac{c_{\mathrm{LB}}}{\sqrt{T}}\textit{)}.$$

**Proof of Claim E.3.** Let $c(A_1), c(A_2)$ denote the expected consumption of arms $A_1$ and $A_2$ respectively in instance $\mathcal{I}$. Define $\zeta := \frac{\epsilon c(A_1)}{1 - c_{\mathrm{LB}}}$. By definition, this implies that the expected consumption of arm $A_2$ in instance $\mathcal{I}'$ is $c(A_2) - \zeta$. Additionally, since the support contains two arms, we have that the following holds: $c(A_1)X_{\mathcal{I}}^*(A_1) + c(A_2)X_{\mathcal{I}}^*(A_2) = B/T * (1 - \eta_{\mathrm{LP}})$ and $c(A_1)X_{\mathcal{I}'}^*(A_1) + c(A_2)X_{\mathcal{I}'}^*(A_2) - \zeta X_{\mathcal{I}'}^*(A_2) = B/T * (1 - \eta_{\mathrm{LP}})$. Thus, we have

$$c(A_1)X_{\mathcal{I}}^*(A_1) + c(A_2)X_{\mathcal{I}}^*(A_2) = c(A_1)X_{\mathcal{I}}^*(A_1) + c(A_2)X_{\mathcal{I}}^*(A_2) + \delta(C(A_2) - c(A_1) - \zeta) - \zeta X_{\mathcal{I}}^*(A_2).$$

Rearranging and using the assumptions in 3.8, we get that

$$\delta = \frac{\zeta X_{\mathcal{I}}^*(A_2)}{c(A_2) - c(A_1) - \zeta} \geq \frac{\epsilon c_{\mathrm{LB}}}{1 - c_{\mathrm{LB}}} \cdot \frac{c_{\mathrm{LB}}}{1 - 2c_{\mathrm{LB}} - \frac{\epsilon \cdot c_{\mathrm{LB}}}{1 - c_{\mathrm{LB}}}} \geq \epsilon \cdot c_{\mathrm{LB}}^2. \qquad (\text{E.4})$$

Consider $\|\boldsymbol{X}_{\mathcal{I}'}^* - \boldsymbol{Y}\|$. This can be rewritten as

$$= \|\boldsymbol{X}_{\mathcal{I}'}^* - \boldsymbol{Y} - \boldsymbol{X}_{\mathcal{I}}^* + \boldsymbol{X}_{\mathcal{I}}^*\|$$

$$\geq \left|\|\boldsymbol{X}_{\mathcal{I}'}^* - \boldsymbol{X}_{\mathcal{I}}^*\| - \|\boldsymbol{X}_{\mathcal{I}}^* - \boldsymbol{Y}\|\right| \qquad \textit{(Triangle inequality)}$$

$$\geq 2\delta - \epsilon \cdot c_{\mathrm{LB}}^2 \qquad \textit{(Premise of the claim and Eq. (E.1))}$$

$$\geq \epsilon \cdot c_{\mathrm{LB}}^2. \qquad \textit{(From Eq. (E.4))}$$

**Proof of Claim E.4.** We will prove the statement $\|\boldsymbol{X}_{\mathcal{I}}^* - \boldsymbol{Y}\| \geq \epsilon \cdot c_{\mathrm{LB}}^2 \implies V_{\mathrm{sc}}(\boldsymbol{X}_{\mathcal{I}}^*) - V_{\mathrm{sc}}(\boldsymbol{Y}) \geq \epsilon \cdot \frac{c_{\mathrm{LB}}^3}{2}$. The exact same argument holds by replacing $\boldsymbol{X}_{\mathcal{I}}^*$ with $\boldsymbol{X}_{\mathcal{I}'}^*$ and $V_{\mathrm{sc}}(.)$ with $V_{\mathrm{sc}}'(.)$.

Consider $V_{\tt sc}(\boldsymbol{X}^*_{\mathcal{I}}) - V_{\tt sc}(\boldsymbol{Y})$. By definition, this equals,

$$r(\boldsymbol{X}^*_{\mathcal{I}}) - \frac{r(\boldsymbol{Y})}{\max\{\frac{B'}{T}, c(\boldsymbol{Y})\}} \cdot \frac{B'}{T}, \tag{E.5}$$

where $B'$ is the scaled budget.

We have two cases. In case 1, let $\max\{\frac{B'}{T}, c(\boldsymbol{Y})\} = \frac{B'}{T}$. Thus, Eq. (E.5) simplifies to,

$$= r(\boldsymbol{X}^*_{\mathcal{I}}) - r(\boldsymbol{Y})$$
$$= r(A_1)[X^*_{\mathcal{I}}(A_1) - Y(A_1)] + r(A_2)[X^*_{\mathcal{I}}(A_2) - Y(A_2)]$$

Note that since $\max\{\frac{B'}{T}, c(\boldsymbol{Y})\} = \frac{B'}{T}$, this implies that $Y(\tt null) = 0$. Since $\boldsymbol{X}^*_{\mathcal{I}}$ is an optimal solution and $r(A_2) > r(A_1)$, this implies that we have $Y(A_1) = X^*_{\mathcal{I}}(A_1) + \zeta$ and $Y(A_2) = X^*_{\mathcal{I}}(A_2) - \zeta$. Thus, we have,

$$r(A_1)[X^*_{\mathcal{I}}(A_1) - Y(A_1)] + r(A_2)[X^*_{\mathcal{I}}(A_2) - Y(A_2)] \geq [r(A_2) - r(A_1)]\zeta$$
$$\geq c_{\tt LB} \cdot \|\boldsymbol{X}^*_{\mathcal{I}} - \boldsymbol{Y}\|/2$$
$$\geq \epsilon \cdot \frac{c_{\tt LB}^3}{2}.$$

Consider case 2 where $\max\{\frac{B'}{T}, c(\boldsymbol{Y})\} = c(\boldsymbol{Y})$. Then, Eq. (E.5) simplifies to,

$$= r(\boldsymbol{X}^*_{\mathcal{I}}) - \frac{B'}{T} \cdot \frac{r(\boldsymbol{Y})}{c(\boldsymbol{Y})}$$
$$\geq r(\boldsymbol{X}^*_{\mathcal{I}}) - \max_{\boldsymbol{Y} \in \Delta_3 : \|\boldsymbol{X}^*_{\mathcal{I}} - \boldsymbol{Y}\| \geq \epsilon \cdot c_{\tt LB}^2} \frac{B(1-\eta_{\tt LP})}{T} \cdot \frac{r(\boldsymbol{Y})}{c(\boldsymbol{Y})}$$

The maximization happens when the distribution $\boldsymbol{Y}$ is such that $Y(A_1) = X^*_{\mathcal{I}} - \epsilon \cdot c_{\tt LB}^2/2$ and $Y(A_2) = X^*_{\mathcal{I}} - \epsilon \cdot c_{\tt LB}^2/2$. Plugging this into the expression we get the RHS is at least,

$$\geq r(\boldsymbol{X}^*_{\mathcal{I}}) - \frac{B(1-\eta_{\tt LP})}{T} \cdot \frac{r(\boldsymbol{X}^*_{\mathcal{I}}) + \epsilon \cdot c_{\tt LB}^2/2 \cdot (r(A_2) - r(A_1))}{c(\boldsymbol{X}^*_{\mathcal{I}}) + \epsilon \cdot c_{\tt LB}^2/2 \cdot (c(A_2) - c(A_1))}$$
$$\geq r(\boldsymbol{X}^*_{\mathcal{I}}) - c_{\tt LB}(1-\eta_{\tt LP}) \cdot \frac{r(\boldsymbol{X}^*_{\mathcal{I}}) + \epsilon \cdot c_{\tt LB}^2/2 \cdot (r(A_2) - r(A_1))}{c(\boldsymbol{X}^*_{\mathcal{I}}) + \epsilon \cdot c_{\tt LB}^2/2 \cdot (c(A_2) - c(A_1))}$$
$$\geq r(\boldsymbol{X}^*_{\mathcal{I}}) - (1-\eta_{\tt LP}) \cdot \frac{r(\boldsymbol{X}^*_{\mathcal{I}}) + \epsilon \cdot c_{\tt LB}^2/2 \cdot (r(A_2) - r(A_1))}{1 + \epsilon \cdot c_{\tt LB}^2/2}$$
$$\geq \frac{\eta_{\tt LP}}{2} \cdot r(\boldsymbol{X}^*_{\mathcal{I}}) \geq \epsilon \cdot \frac{c_{\tt LB}^3}{2}.$$

The last two inequality follows from Assumption 3.8-(3.8), the value of $\eta_{\tt LP}$ and the fact that $\epsilon = \frac{c_{\tt LB}}{\sqrt{T}}$, respectively. Combining the two cases we get the claim.

## F   Proof of Theorem 3.9(b): $\sqrt{T}$ lower bound for $d > 2$

We first show that for any given instance $\mathcal{I}_0$, for a given $0 < \delta_1 \leq \mathcal{O}\left(\frac{1}{\sqrt{T}}\right)$ we can obtain a $\delta_1$-perturbation of this instance, denoted by $\mathcal{I}_0'$, that satisfies Eq. (3.15). Given instance $\mathcal{I}_0$ we construct the $\delta_1$-perturbation as follows. We construct instance $\mathcal{I}_0'$ by decreasing the mean consumption on arm $A_i$ and resource $j$ by $\zeta_1^j$. We keep the mean rewards the same. Let $\boldsymbol{X}$ denote the optimal solution to instance $\mathcal{I}$. As a notation we denote the matrix $\boldsymbol{C} \in [0,1]^{d \times 3}$ as the matrix of mean consumption. Let $\boldsymbol{B}$ denote the sub-matrix of $\boldsymbol{C}$ such that, $\boldsymbol{X}$ satisfies the constraints in the scaled LP (C.1) with equality. Thus, we have $\boldsymbol{C} \cdot \boldsymbol{X} = \boldsymbol{b}$, where every co-ordinate of $b$ is $\frac{B(1-\eta_{\tt LP})}{T}$. Thus, the perturbation is equivalent to perturbing the vector $\boldsymbol{b}$, such that the $j^{th}$ entry has an additive perturbation of $\zeta^j$. From Proposition 3.1 in [46], this linear program has a non-degenerate primal optimal solution, in the sense that it satisfies Eq. (3.15).

Next, we show that given an instance $\mathcal{I}_0'$ we can obtain a $\delta_2$ perturbation of $\mathcal{I}_0'$ for a given $0 < \delta_2 \leq \mathcal{O}\left(\frac{1}{\sqrt{T}}\right)$, such that the consumption vectors are linearly independent. Define a random matrix

$D \in [-\zeta_2, \zeta_2]^{d \times 3}$ such that every entry in $D$ is generated uniformly at random from the set $[-\zeta_2, \zeta_2]$. We claim that the vectors $c_j - d_j$ are all linearly independent, where $d_j$ is the $j^{th}$ row of $D$ with probability at least $0.6$. In other words, decreasing each of the mean consumption by a uniformly random value chosen from the set $[-\zeta_2, \zeta_2]$ implies that there exists a realization of $D$ such that the vectors $c_j - d_j$ are all linearly independent.

The proof of this claim proceeds as follows. As before define $C \in [0,1]^{d \times 3}$ to be the matrix of mean consumption. From definition of linear independence we need to show that the smallest singular value of the matrix $C - D$ is non-zero. Note that every entry in the matrix $C - D$ is chosen independently. Thus, using the bound on the probability of singularity in Theorem 2.2 of [21] we have that the probability that the smallest singular value is 0 is at most $\frac{1}{2\sqrt{2}}$. Thus, with probability at least $1 - \frac{1}{2\sqrt{2}} > 0.6$ we have that the matrix $C - D$ is singular.

Thus, for $\delta := \delta_1 + \delta_2$, we have that there exists a $\delta$-perturbed instance $\widetilde{\mathcal{I}}_0$, that satisfies all the assumptions in 3.8 and linear independence condition required in the premise of Theorem 3.10.

# G   Simple regret: proof of Theorem 4.1

For convenience, let us restate the theorem:

**Theorem.** *Assume $B \geq \Omega(T)$ and $\eta_{\mathrm{LP}} \leq \frac{1}{2}$. With probability at least $1 - O(T^{-3})$, for each $\epsilon > 0$, there are at most $N_\epsilon = \mathcal{O}\left(\frac{K}{\epsilon^2} \log KTd\right)$ rounds $t$ such that $\mathsf{OPT}_{\mathsf{DP}}/T - r(X_t) \geq \epsilon$.*

The proof consists of two major steps: we argue about confidence sums, and we upper-bound simple regret in terms of the confidence radius.

## G.1   Confidence sums

The following arguments depend only on the definition of the confidence radius, and work for any algorithm `ALG`. Suppose in each round $t$, this algorithm chooses a distribution $Y_t$ over arms and samples arm $a_t$ independently $Y_t$. We upper-bound the number of rounds $t$ with large $\mathrm{Rad}_t(Y_t)$:

**Lemma G.1.** *Fix the threshold $\theta_0 > 0$, and let $S$ be the set of all rounds $t \in [T]$ such that $\mathrm{Rad}_t(Y_t) \geq \theta_0$. Then $|S| \leq \mathcal{O}\left(\theta_0^{-2} \cdot K \log(KdT)\right)$ with probability at least $1 - O(T^{-3})$.*

To prove the lemma, we study *confidence sums*: for a subset $S \subset [T]$ of rounds, define

$$W_{\mathtt{act}}(S) := \sum_{t \in S} \mathrm{Rad}_t(a_t) \qquad \textit{(action-confidence sum of ALG)},$$
$$W_{\mathtt{dis}}(S) := \sum_{t \in S} \mathrm{Rad}_t(Y_t) \qquad \textit{(distribution-confidence sum of ALG)}.$$

First, a standard argument (*e.g.,* implicit in [10], see Section G.4) implies that

$$W_{\mathtt{act}}(S) \leq \mathcal{O}\left(\sqrt{K |S| C_{\mathtt{rad}}} + K \cdot \ln |S| \cdot C_{\mathtt{rad}}\right) \quad \text{for any fixed subset } S \subset [T]. \qquad \text{(G.1)}$$

Second, note that $W_{\mathtt{dis}}(S)$ is close to $W_{\mathtt{act}}(S)$: for any fixed subset $S \subset [T]$,

$$|W_{\mathtt{dis}}(S) - W_{\mathtt{act}}(S)| \leq \mathcal{O}(\sqrt{|S| \log T}) \quad \text{with probability at least } 1 - T^{-3}. \qquad \text{(G.2)}$$

This is by Azuma-Hoeffding inequality, since $\left(\mathrm{Rad}_t(a_t) - \mathrm{Rad}_t(Y_t) : t \in S\right)$ is a martingale difference sequence. We extend this observation to *random* sets $S$. A random set $S \subset [T]$ is called *time-consistent* if the event $\{t \in S\}$ does not depend on the choice of arm $a_t$ or anything that happens afterwards, for each round $t$. (But it *can* depend on the choice of distribution $Y_t$.)

**Claim G.2.** *For any any time-consistent random set $S \subset [T]$,*

$$|W_{\mathtt{dis}}(S) - W_{\mathtt{act}}(S)| \leq \mathcal{O}\left(\sqrt{|S| \log T} + \log T\right) \quad \text{with probability at least } 1 - T^{-3}. \qquad \text{(G.3)}$$

*Proof.* By definition of time-consistent set, for each round $t$,

$$\mathbb{E}[\mathbf{1}_{\{t \in S\}} \cdot \mathrm{Rad}_t(a_t) \mid (Y_1, a_1), \ldots, (Y_{t-1}, a_{t-1}), Y_t] = \mathbf{1}_{\{t \in S\}} \cdot \mathrm{Rad}_t(Y_t).$$

Thus, $\mathbf{1}_{\{t \in S\}} \mathrm{Rad}_t(a_t) - \mathrm{Rad}_t(Y_t), t \in [T]$ is martingale difference sequence. Claim G.2 follows from a concentration bound from prior work (Theorem D.3). $\qquad \square$

We complete the proof of Lemma G.1 as follows. Fix $\delta > 0$. Since $S$ is a time-consistent random subset of $[T]$, by Eq. (G.1) and Claim G.2, with probability at least $1 - \delta$ it holds that

$$\theta_0 \cdot |S| \leq W_{\texttt{dis}}(S) \leq \mathcal{O}\left(\sqrt{|S|KC_{\texttt{rad}}} + K\,C_{\texttt{rad}} + \sqrt{|S|\,\log T} + \log T\right).$$

We obtain the Lemma by simplifying and solving this inequality for $|S|$.

## G.2 Connecting LP-gap and the confidence radius

In what follows, let $B_{\texttt{sc}} = B(1 - \eta_{\texttt{LP}})$ be the budget in the rescaled LP.

**Lemma G.3.** *Fix round $t \in [T]$, and assume the "clean event" in (2.7). Then*

$$G_{LP}(\boldsymbol{X}_t) \leq (2 + {}^T\!/_{B_{\texttt{sc}}})\,\text{Rad}_t(\boldsymbol{X}_t).$$

*Proof.* Let $\alpha := B_{\texttt{sc}}/T$. For any distribution $\boldsymbol{X}$, let

$$V_+(\boldsymbol{X}) := {}^{B_{\texttt{sc}}}\!/_T \;\cdot r(\boldsymbol{X})/\max_{j \in [d]} c_j^-(\boldsymbol{X}).$$

denote the value of $\boldsymbol{X}$ in the optimistic LP (2.6), after proper rescaling. Let $\boldsymbol{X}^*$ be an optimal solution to the (original) LP (2.2). Then

$$G_{\text{LP}}(\boldsymbol{X}_t) = V(\boldsymbol{X}^*) - V(\boldsymbol{X}_t) - V_+(\boldsymbol{X}_t) + V_+(\boldsymbol{X}_t). \tag{G.4}$$

Since $V_+(\boldsymbol{X}_t)$ is the optimal solution to the optimistic LP (2.6),

$$V_+(\boldsymbol{X}_t) \geq V_+(\boldsymbol{X}^*).$$

Moreover, since $\boldsymbol{X}^*$ is feasible to the optimistic LP (2.6) with the scaled budget $B_{\texttt{sc}}$,

$$V_+(\boldsymbol{X}^*) \geq V(\boldsymbol{X}^*).$$

It follows that Eq. (G.4) an be upper-bounded as

$$G_{\text{LP}}(\boldsymbol{X}_t) \leq V_+(\boldsymbol{X}_t) - V(\boldsymbol{X}_t). \tag{G.5}$$

We will now upper-bound the right-hand side in the above. Denote

$$c_{\max}(\boldsymbol{X}_t) := \max_{j \in [d]} \sum_{a \in [K]} c_{j,t}(a) X_t(a)$$

$$c_{\max}^-(\boldsymbol{X}_t) := \max_{j \in [d]} \sum_{a \in [K]} c_{j,t}^-(a) X_t(a).$$

By definition of the value of a linear program, we can continue Eq. (G.5) as follows:

$$G_{\text{LP}}(\boldsymbol{X}_t) \leq V_+(\boldsymbol{X}_t) - V(\boldsymbol{X}_t)$$
$$\leq \alpha \cdot \frac{\hat{r}(\boldsymbol{X}_t) + \text{Rad}_t(\boldsymbol{X}_t)}{c_{\max}^-(\boldsymbol{X}_t)} - \alpha \cdot \frac{r(\boldsymbol{X}_t)}{c_{\max}(\boldsymbol{X}_t)}. \tag{G.6}$$

Under the clean event in Eq. (2.7), we continue Eq. (G.6) as follows:

$$\leq \alpha \left(\frac{2\,\text{Rad}_t(\boldsymbol{X}_t) + r(\boldsymbol{X}_t)}{c_{\max}^-(\boldsymbol{X}_t)} - \frac{r(\boldsymbol{X}_t)}{c_{\max}(\boldsymbol{X}_t)}\right). \tag{G.7}$$

Since time is one of the resources, $c_{\max}^-(\boldsymbol{X}_t) \geq \frac{B_{\texttt{sc}}}{T}$. Thus, we continue Eq. (G.7) as follows:

$$\leq 2\,\text{Rad}_t(\boldsymbol{X}_t) + \alpha r(\boldsymbol{X}_t)\left(\frac{1}{c_{\max}^-(\boldsymbol{X}_t)} - \frac{1}{c_{\max}(\boldsymbol{X}_t)}\right)$$

$$= 2\,\text{Rad}_t(\boldsymbol{X}_t) + \alpha r(\boldsymbol{X}_t)\left(\frac{\text{Rad}_t(\boldsymbol{X}_t)}{c_{\max}^-(\boldsymbol{X}_t) \cdot c_{\max}(\boldsymbol{X}_t)}\right)$$

$$\leq 2\,\text{Rad}_t(\boldsymbol{X}_t) + \frac{\text{Rad}_t(\boldsymbol{X}_t)}{c_{\max}^-(\boldsymbol{X}_t)} \tag{G.8}$$

$$\leq \left(2 + \frac{T}{B_{\texttt{sc}}}\right)\text{Rad}_t(\boldsymbol{X}_t) \tag{G.9}$$

Eq. (G.8) uses the fact that $\alpha \frac{r(\boldsymbol{X}_t)}{c_{\max}(\boldsymbol{X}_t)} \leq \frac{B}{T}\frac{r(\boldsymbol{X}_t)}{c_{\max}(\boldsymbol{X}_t)} = V(\boldsymbol{X}_t) \leq 1$. Eq. (G.9) uses the fact that time is one of the resources and thus, $c_{\max}^-(\boldsymbol{X}_t) \geq \frac{B_{\texttt{sc}}}{T}$. $\qquad\square$

### G.3 Finishing the proof of Theorem 4.1

**Claim G.4.** *Fix round $t$, and assume the "clean event" in (2.7). Then*

$$\mathtt{OPT}_{DP}/T - r(\boldsymbol{X}_t) \le G_{LP}(\boldsymbol{X}_t) + \eta_{\mathrm{LP}}.$$

*Proof.* By (2.7) and because $\boldsymbol{X}_t$ is the solution to the optimistic LP, we have

$$\max_{j \in d} c_j(\boldsymbol{X}_t) \ge \max_{j \in d} c_j^-(\boldsymbol{X}_t) = {}^{B}/_{T}\,(1 - \eta_{\mathrm{LP}}).$$

It follows that $r(\boldsymbol{X}_t) \ge V(\boldsymbol{X}_t)(1 - \eta_{\mathrm{LP}})$. Finally, we know that $\mathtt{OPT}_{\mathrm{LP}} \ge \mathtt{OPT}_{\mathrm{DP}}/T$. $\square$

Condition on (2.7), and the high-probability event in Lemma G.1. (Take the union bound in Lemma G.1 over all thresholds $\theta_0 \ge 1/\sqrt{T}$, *e.g.,* over an exponential scale.) Fix $\epsilon > 0$. By Claim G.4 and Lemma G.3, any round $t$ with simple regret at least $\epsilon$ satisfies

$$\epsilon \le \mathtt{OPT}_{\mathrm{DP}}/T - r(\boldsymbol{X}_t) \le \eta_{\mathrm{LP}} + (2 + {}^{T}/_{B_{\mathrm{sc}}})\,\mathrm{Rad}_t(\boldsymbol{X}_t).$$

Therefore, $\mathrm{Rad}_t(\boldsymbol{X}_t) \ge \theta_0$, where $\theta_0 = \frac{\epsilon - \eta_{\mathrm{LP}}}{(2 + T/B_{\mathrm{sc}})} \ge \Theta(\epsilon)$ when $\epsilon \ge 2\eta_{\mathrm{LP}}$. Now, the theorem follows from Lemma G.1. Note, when $\epsilon < 2\eta_{\mathrm{LP}}$, then the total number of rounds in the theorem is larger than $T$ and hence not meaningful.

### G.4 The standard confidence-sum bound: proof of Eq. (G.1)

Let us prove Eq. (G.1) for the sake of completeness. By definition of $\mathrm{Rad}_t(a_t)$ from Eq. (2.8),

$$\mathrm{Rad}_t(a_t) = f(n) := \min\left(1,\ \sqrt{C_{\mathtt{rad}}/n} + C_{\mathtt{rad}}/n\right),$$

where $N_t(a)$ is the number of times arm $a$ was chosen before round $t$. Therefore:

$$
\begin{aligned}
\sum_{t \in S} \mathrm{Rad}_t(a_t) &\le \sum_{a \in [K]} \sum_{n=1}^{|S|/K} f(n) \\
&\le \sum_{a \in [K]} \int_{x=1}^{|S|/K} f(x)\,\mathrm{d}x \le 3\left(\sqrt{K|S|\,C_{\mathtt{rad}}} + K \cdot \ln|S| \cdot C_{\mathtt{rad}}\right).
\end{aligned}
$$

## H  Reduction from BwK to bandits

We extend our results to any problem which can be cast as a special case of BwK and admits an upper bound on action-confidence sums, in the style of (G.1), for a suitably defined confidence radius.

To state the general result, let us define an abstract notion of "confidence radius". For each round $t$, a *formal confidence radius* is a mapping $\mathrm{Rad}_t(a)$ from algorithm's history and arm $a$ to $[0, 1]$ such that with probability at least $1 - O(T^{-4})$ it holds that

$$|r(a) - \hat{r}_t(a)| \le \mathrm{Rad}_t(a) \quad \text{and} \quad |c_j(a) - \hat{c}_{j,t}(a)| \le \mathrm{Rad}_t(a)$$

for each resource $j$, where $\hat{r}_t(a)$ and $\hat{c}_{j,t}(a)$ denote average reward and resource consumption, as defined in Eq. (B.3). Such $\mathrm{Rad}_t(a)$ induces a version of UcbBwK with confidence bounds

$$r_t^+(a) = \min(1, \hat{r}_t(a) + \mathrm{Rad}_t(a)\ ) \quad \text{and} \quad c_{j,t}^-(a) = \max(\,0, \hat{c}_{j,t}(a) - \mathrm{Rad}_t(a)\ ).$$

We allow the algorithm to observe auxiliary feedback before and/or after each round, depending on a particular problem formulation, and this feedback may be used to compute the confidence radii.

We replace Eq. (G.1) with a generic bound on the action-confidence sum, for some $\beta$ that can depend on the parameters in the problem instance, but not on $S$:

$$\sum_{t \in S} \mathrm{Rad}_t(a_t) \le \sqrt{|S|\,\beta}, \quad \text{for any algorithm and any subset } S \subset [T]. \tag{H.1}$$

**Theorem H.1.** *Consider an instance of BwK with time horizon $T$. Let $\mathrm{Rad}_t(\cdot)$ be a formal confidence radius which satisfies* (H.1) *for some $\beta$. Consider the induced algorithms* UcbBwK *and* PrunedUcbBwK *with rescaling parameter $\eta_{\mathrm{LP}} = \frac{2}{B}\sqrt{\beta T}$.*

*(i) Both algorithms obtain regret $\mathtt{OPT_{DP}} - \mathbb{E}[\mathtt{REW}] \leq O(\sqrt{\beta T})(1 + \mathtt{OPT_{DP}}/B)$.*
*(ii) Theorem 3.2 holds with $\Psi = \beta\,G_{LAG}^{-2}$ and regret $\mathcal{O}\left(\beta\,G_{LAG}^{-1}\right)$ in part (ii).*
*(iii) Theorem 4.1 holds with $N_\epsilon = \mathcal{O}\left(\beta\,\epsilon^{-2}\right)$.*

**Proof Sketch** For part (i), the analysis in [3] explicitly relies on (G.1). For part (ii), we modify the proof of Theorem 3.2 so as to use (G.1) instead of Claim 3.4. For part (iii), our proof of Theorem 4.1 uses (G.1) explicitly. In all three parts, we replace (G.1) with (H.1), and trace how the latter propagates through the respective proof. ∎

We apply this general result to three specific scenarios: linear contextual bandits with knapsacks (LinCBwK) [5], combinatorial semi-bandits with knapsacks (SemiBwK) [49], and multinomial-logit bandits with knapsacks (MnlBwK) [26]. In all three applications, the confidence-sum bound (H.1) is implicit in prior work on the respective problem without resources. The guarantees in part (i) match those in prior work referenced above, up to logarithmic factors, and are optimal when $B = \Omega(T)$; in fact, we obtain an improvement for MnlBwK. Parts (ii) and (iii) – the results for logarithmic regret and simple regret – did not appear in prior work.

## H.1 Linear Contextual Bandits with Knapsacks (LinCBwK)

In *Contextual Bandits with Knapsacks* (CBwK), we have $K$ actions, $d$ resources, budget $B$ and time horizon $T$, like in BwK, and moreover we have a set $\mathcal{X}$ of possible contexts. At each round $t \in [T]$, the algorithm first obtains a context $\boldsymbol{x}_t \in X$. The algorithm then chooses an action $a_t \in [K]$ and obtains an outcome $\boldsymbol{o}_t(a_t) \in [0,1]^{d+1}$ like in BwK. The tuple $(\boldsymbol{x}_t; \boldsymbol{o}_t(a) : a \in [K])$ is drawn independently from some fixed but unknown distribution. The algorithm continues until some resource, including time, is exhausted. One compares against a given a set $\Pi$ of *policies*: mappings from contexts to actions. We can formally interpret CBwK as an instance of BwK in which actions correspond to policies in $\Pi$. This interpretation defines the benchmarks $\mathtt{OPT_{DP}}$ and $\mathtt{OPT_{FD}}$ that we compete with.

LinCBwK is a special case of CBwK in which the context space is $\mathcal{X} = [0,1]^{K \times m}$, for some parameter $m \in \mathbb{N}$, so that each context $\boldsymbol{x}_t$ is in fact a tuple $\boldsymbol{x}_t = (\boldsymbol{x}_t(a) \in [0,1]^m : a \in [K])$. We have a linearity assumption: for some unknown matrix $\boldsymbol{W}_* \in [0,1]^{m \times (d+1)}$ and each arm $a \in [K]$,

$$\mathbb{E}\left[\boldsymbol{o}_t(a) \mid \boldsymbol{x}_t(a)\right] = \boldsymbol{W}_*^{\mathsf{T}} \cdot \boldsymbol{x}_t(a).$$

The policy set $\Pi$ consists of all possible policies.

*Linear contextual bandits*, studied in prior work [*e.g.,* 9, 29, 43, 27, 2], is the special case without resources. Much of the complexity of linear contextual bandits (resp., LinCBwK) is captured by the special case of of *linear bandits* (resp., *linear BwK*) where the context is the same in each round.

The general theme in the work on linear bandits (contextual or not) to replace the dependence on the number of arms $K$ in the regret bound with the dependence on the dimension $m$ and, if applicable, avoid the dependence on $|\Pi|$. This is what we accomplish, too.

**Corollary H.2.** *For LinCBwK, Theorem H.1 holds with $\beta = \mathcal{O}(m^2 d^2 \log(mTd))$.*

*Proof.* Combining Lemma 13 of [9] and Theorem 2 of [1], it follows that the confidence-sum bound Eq. (H.1) holds with $\beta = \mathcal{O}(m^2 d^2 \log mTd)$. □

## H.2 Combinatorial Semi-bandits with Knapsacks (SemiBwK)

SemiBwK is a version of BwK, where actions correspond to subsets of some fixed ground set $[N]$ (whose elements are called *atoms*). There is a fixed family $\mathcal{F} \subset 2^{[N]}$ of feasible actions. In each round $t$, the algorithm chooses a subset $A_t \in \mathcal{F}$ and observes the outcome $\boldsymbol{o}_t(a) \in [0, 1/n]^d$ for each atom $a \in A_t$, where $n = \max_{A \in \mathcal{F}} |A|$. The outcome for a given subset $A \in \mathcal{F}$ is defined as the sum

$$\boldsymbol{o}_t(A) = \sum_{a \in A} \boldsymbol{o}_t(a) \in [0,1]^{d+1}. \tag{H.2}$$

The outcome matrix $(\boldsymbol{o}_t(a) : a \in [N])$ is drawn independently from some fixed but unknown distribution. The algorithm continues until some resource, including time, is exhausted.

*Combinatorial semi-bandits*, the problem studied in prior work [*e.g.,* 25, 40, 39], is the special case without resources. Note that the number of feasible actions can be exponential in $N$. The general

theme in this line of work is to replace the dependence on $|\mathcal{F}|$ in the regret bound with the dependence on $N$, or, even better, on $n$. We extend this to `SemiBwK`.

**Corollary H.3.** *For* `SemiBwK`, *Theorem H.1 holds with* $\beta = \mathcal{O}(n \log(N d T))$.

*Proof.* Using Lemma 4 in [60] we immediately obtain the confidence-sum bound Eq. (H.1) with $\beta = n \log K d T$. $\qquad\square$

### H.3 Multinomial-logit Bandits with Knapsacks (`MnlBwK`)

In the `MnlBwK` problem, the setup starts like in `SemiBwK`. There is a ground set of $N$ *atoms*, and a fixed family $\mathcal{F} \subset 2^{[N]}$ of feasible actions. In each round, each atom $a$ has an outcome $\boldsymbol{o}_t(a) \in [0,1]^{d+1}$, and the outcome matrix $(\boldsymbol{o}_t(a) : a \in [N])$ is drawn independently from some fixed but unknown distribution. The aggregate outcome is formed in a different way: when a given subset $A_t \in \mathcal{F}$ is chosen by the algorithm in a given round $t$, at most one atom $a_t \in A_t$ is chosen stochastically by "nature", and the aggregate outcome is then $\boldsymbol{o}_t(A_t) := \boldsymbol{o}_t(a)$; otherwise, the algorithm skips this round. A common interpretation is that the atoms correspond to products, the chosen action $A_t \in \mathcal{F}$ is the bundle of products offered to the customer, and at most one product from this bundle is actually purchased. As usual, the algorithm continues until some resource (incl. time) is exhausted.

The selection probabilities are defined via the multinomial-logit model. For each atom $a$ there is a hidden number $v_a \in [0,1]$, interpreted as the customers' valuation of the respective product, and the

$$\Pr\left[\,\text{atom } a \text{ is chosen} \mid A_t\,\right] = \begin{cases} \frac{v_a}{1 + \sum_{a' \in A_t} v_{a'}} & \text{if } a \in A_t \\ 0 & \text{otherwise.} \end{cases}$$

The set $\mathcal{F}$ of possible bundles is

$$\mathcal{F} = \{\, A \subset [N] : \ \boldsymbol{M} \cdot x(A) \leq \boldsymbol{b} \,\},$$

for some (known) totally unimodular matrix $\boldsymbol{M} \in \mathbb{R}^{N \times N}$ and a vector $\boldsymbol{b} \in \mathbb{R}^N$, where $x(A) \in \{0,1\}^N$ represents set $A$ as a binary vector over atoms.

*Multinomial-logit bandits*, the problem studied in prior work [*e.g.,* 7, 48, 51, 24], is the special case without resources. We derive the following corollary from the analysis of MNL-bandits in Agrawal et al. [7], which analyzes the confidence sum for the $v_a$'s.

**Corollary H.4.** *Consider* `MnlBwK` *and denote* $V := \sum_{a \in [N]} v_a$. *Theorem H.1 holds with*

$$\beta = \mathcal{O}\left(\left(\frac{\ln T}{\ln(1 + 1/V)}\right)^2 \left(N\sqrt{\ln(NT)} + \ln(NT)\right)\right) = \widetilde{O}\left(N^3\right).$$

*Proof.* The proof is implicit in the analysis in Agrawal et al. [7]. As in their paper, let $n_\ell$ denote the number of time-steps in phase $\ell$. Let $V_\ell = \sum_{a \in S_\ell} v_a$. Recall that $n_\ell$ is a geometric random variable with mean $\frac{1}{1 + V_\ell}$. Using Chernoff-Hoeffding bounds we obtain that with probability at least $1 - \frac{1}{T^2}$, $n_\ell \leq \frac{\ln T}{\ln(1 + 1/V_\ell)}$.

Consider a random subset $S$. Summing the LHS and RHS in Lemma 4.3, we get that $\sum_{t \in S} \text{Rad}_t(a_t) \leq \sum_{a \in [N]} \sum_{\ell : t \in \mathcal{T}_a(\ell)} \tilde{R}_a(S_\ell)$. Using Lemma 4.3 in [7] we have, $\sum_{a \in [N]} \sum_{\ell : t \in \mathcal{T}_a(\ell)} \tilde{R}_a(S_\ell) \leq \sum_{a \in [N]} \sum_{\ell : t \in \mathcal{T}_a(\ell)} n_\ell \sqrt{\frac{v_a \ln \sqrt{N}T}{T_a(\ell)}} + \frac{\ln \sqrt{N}T}{T_a(\ell)}$. Note that $v_a \leq 1$. Using the upper bound on $n_\ell$ derived above combined with the argument used to obtain (A.19) in [7] we get the desired value of $\beta$. $\qquad\square$

The worst-case regret bound from Corollary H.4 improves over prior work [26]. In particular, consider the worst-case dependence on $N$, the number of atoms. Our regret bound scales as $N^{3/2}$, whereas the regret bound in [26] scales as $N^{7/2}$ (while both scale as $\sqrt{T}$).

## H.4  Computational issues

We do not provide a generic computationally efficient implementation for `UcbBwK` in our reduction. The algorithm constructs and solves a linear program in each round, with one variable per arm in the reduction. So, even if the regret is fairly small, the number of LP variables may be very large: indeed, it may be exponential in the number of atoms in `SemiBwK` and `MnlBwK`, arbitrarily large compared to the other parameters in linear `BwK`, or even infinite as in `LinCBwK`. The corresponding LPs have a succinct representation in all these applications, but we do not provide a generic implementation. However, such (or very similar) linear programs may be computationally tractable via application-specific implementations, and indeed this is the case in `LinCBwK` [5] and `SemiBwK` [49]. In the prior work on `MnlBwK` [26], the $\sqrt{T}$-regret algorithm is not computationally efficient, same as ours; there is, however, a computationally efficient algorithm with regret $T^{2/3}$.