# OpenReview forum: "Bandits with Knapsacks beyond the Worst Case"
_NeurIPS.cc/2021/Conference — NeurIPS 2021 Poster_

### Official Review · Reviewer_3DAX · 2021-07-15

**Rating:** 6
**Confidence:** 4

**Summary:**

The paper introduces a new algorithm for the BwK Problem achieving $O(\log T)$ regret for a "best-arm-optimal" instance. The algorithm is based on a variant of UcbBwK. Moreover, they show $\Omega(\sqrt{T})$ lower bound and they provide a general reduction from BwK to stochastic bandits.

Pros:
(i) Characterizing the problem-dependent lower bound by a new quantity, Lagrangian gap. In the language of linear programming (LP), this is related to minimal non-zero reduced cost for the underlying LP that does not include the null arm.
(ii) Better bound. From certain "best-arm-optimal" instances, their regret bound is better than the existing work.
(iii) One-round performance. It shows that the rewards collected by UcbBwK are approximately optimal for all but a few rounds.
(iv) Reduction from BwK to stochastic bandits.



**Limitations And Societal Impact:**

There is no negative societal impact that I can see.

**Main Review:**

Given the merits summarized above, I have the following concerns:

The recent work [44] studies the BwK problem under a more general setting without the best-arm-optimal condition. It would be helpful to present a comparison of the bound in this paper and the bound therein (when reduced to a best-arm-optimal instance).

There might be some issues with the lower bound proof:
(i) The second line on the inequalities at the bottom of page 22 might not hold. First, intuitively, for an algorithm that only plays non-null arms at the last few rounds, $V({Y})$ can be larger than $V({X}^*)$ in those rounds since during those rounds, the resources average budget is more than the initial average budget and scaled average budget. Second, theoretically, in the proof of E.4, it seems that authors assume that the policy ${Y}$ at each round must be a feasible solution to the LP corresponding to ${X}^*$ (thus the performance of ${Y}$ can not be better than $V(X^*)$ in that step). However, the average budget in a step can be larger than the initial average budget. Thus, even if ${Y}$ and ${X}^*$ are different, $V({Y})$ can still be larger than $V({X}^*)$ in this case.

(ii) The first paragraph of Appendix F might require more clarifications. First, it seems unclear why the perturbation on $r$ and ${C}$ is equivalent to perturbing the vector ${b}$ in general. More importantly, Proposition 3.1 in [46] only provides a sufficient condition that small perturbation can help a degenerate LP to become non-degenerate. Authors might need more explanation on how the perturbation can render an LP from non-degenerate to degenerate (reversely from [46]).

**Time Spent Reviewing:**

5

---

> ### Author Response · Authors · 2021-08-10
> **Thanks for the positive review and we address all stated issues below.**
>
>
>
> ## Re comparison to [44]
>
> We discuss [44] in Lines 73-76, making these points:
>
> - They provide a logarithmic regret bound without our assumption of "best-arm-optimality", but with several other parameters. These parameters must "blow up" to yield >= sqrt{T} regret whenever our lower bounds apply.
> - While it is possible that their regret bound improves over ours in some regimes, no examples are provided when all their parameters are small, e.g., so that their regret bound is o(sqrt{T}), let alone O(log T).
> - Their algorithm does not achieve o(T) regret in the worst case.
>
>  Conceptually, "best-arm-optimality" is replaced with another assumption: a lower bound on the positive entries of the optimal distribution x*. (This is parameter $\chi$ in Section 3.3 of [44].)  It is unclear to us if this assumption is non-degenerate. (Whereas best-arm-optimality is a non-degenerate property, see Footnote 5 on page 9.)
>
> In fact, the regret bound in [44] yields Omega(T) regret for the specific example in our lower bounds (lines 264-267).  This is because parameter $\chi$ mentioned above is ~ $1/\sqrt{T}$. In fact, parameter $\delta$ in [44] is of the same order, which yields $\Omega(T)$ regret independently of $\chi$.
>
> ## Suspected issues with the lower bound proof
>
> These are all about Appendix E -- thanks for reading our paper in such detail!
>
> ### Issue (i)
>
> To prove this inequality, we use Claim E.4. This claim invokes the specific assumptions that we make about the problem instance in our lower bound. Your stated intuition does not seem to account for these assumptions.
>
> To clarify a few potential misunderstandings:
>
> - Claim E.4 is not about algorithms or executions thereof. It is a purely "algebraic" statement about formulas, which applies to any distribution $Y$ over arms that satisfies some condition. In particular, $Y$ is not assumed to be LP-feasible.
> - Quantity $V(X)$ is a fixed formula for a given distribution $X$ over arms, not something that changes over time in the execution of an algorithm.
> - $V(X)$ is defined in Lines 787-788 as the LP-value of $X$ in the LP (C.1), where the value of $X$ in a given LP is defined in Eq. (4.1). This definition invokes rescaling the distribution when/if needed, to make the distribution LP-feasible. Admittedly, we could spell out these definitions way more clearly!
> - V(X) is well-defined even if distribution X is not LP-feasible. If X is not LP-feasible, then V() rescales it to make it feasible. (Likewise, an algorithm that samples from such X in each round would need to stop early.)
> - The key property of V(X) is that the optimal value of the LP equals sup_{distributions X} V(X). (This is not difficult to see, and also observed, e.g., in Eq. (8) in [14].)
>
> Another subtlety is that our analysis focuses on a rescaled problem instance, as per Lines 756-758.
>
> ### Issue (ii)
>
> The equivalence is because for each row, every entry is perturbed using the same noise term. Thus, a simple algebraic re-arrangement implies that the set of feasible solutions remains the same when an equivalent perturbation on $b$ is applied instead.
>
> We apologize for a typo in Line 836: we mean "non-degenerate" rather than "degenerate" there (in the sense of satisfying Eq. (3.15), as we point out). Thus, we are using Prop. 3.1 in [46] as it is: perturbations make a degenerate LP into a non-degenerate one.

---

### Official Review · Reviewer_9Bz5 · 2021-07-16

**Rating:** 6
**Confidence:** 4

**Summary:**

This paper studies the problem of bandits with knapsacks (BwK). In a BwK instance, there are $K$ arms and multiple types of resources with limited budgets, while in standard bandits there is only one resource time. Each pull of an arm returns a random reward and consumes a random amount of each resource. The pulling ends as soon as one of the resources runs out, and the goal is to maximize the cumulative reward before a resource runs out.

In particular, this paper focuses on the case (denoted as $C$) where there are only two types of resources and the optimal policy is always pulling some arm $a^*$. In this specific case, the authors showed a regret upper bound in the form of $O(\frac{K}{\Delta^2}\log{T})$ or $O(\frac{K}{\Delta}\log{T})$ under some conditions. The authors also showed that if $C$ does not hold, then no algorithm can do better than $\Omega(\sqrt{T})$ and thus, no logarithmic regret can be obtained.

The authors also studied the simple regret of BwK and showed that their algorithm can achieve $O(\log{T})$ simple regret.

**Limitations And Societal Impact:**

See the comments in the box Main Review.

**Main Review:**

The logarithmic regret of BwK has been a difficult problem, and there are only a very few works that have focused on this topic. To the best of my knowledge, this work is the first to obtain a logarithmic regret with i) more than one type of resource and ii) random or non-deterministic regrets. When one of i) or ii) does not hold, then previous works have shown logarithmic regrets, but when both of them are true, then this work is the first. However, I do not find a lot of novelty in the algorithm design and this specific case may be limited in practice.

This paper is not quire well written as there are too many notations. Perhaps the authors can try simplifying or hiding some of the notations in the next version.

In my opinion, the logarithmic lower bound of BwK is also difficult, but it would be better if the authors can provide some results on the lower bound. It is not clear whether $G_{\mbox{LAG}$ is a good or tight enough gap for deriving the regret.

Thus, I will vote a 6 for this submission. This work has significant contribution to a specific case, the contribution is limited in practice and lacks the justification that this result is good enough.

**Time Spent Reviewing:**

2

---

> ### Author Response · Authors · 2021-08-10
> **Thanks for the positive review**
>
> ### Re: "I do not find a lot of novelty in the algorithm design"
>
> Indeed, we do not invent a new algorithm! Instead, we analyze a (very reasonable) existing algorithm with optimal worst-case performance. The novelty lies in (several) new analyses of this algorithm, along with the lower bounds. We believe that not inventing ad-hoc algorithms is a feature rather than a flaw.
>
> ### Re significance:
>
> BwK with >1 resources and randomized resource consumption is, arguably, the main regime of interest. Indeed, as per para -1 of related work:
>
> - All motivating examples for BwK in the literature require resource consumption to be stochastic (and in fact correlated with rewards).
> - Having only 1 constrained resource (and infinite time horizon) makes the problem much easier, and also appears fairly unrealistic.
>
> The significance of our log-regret results is two-fold: (i) we obtain a full answer for a natural question of when log(T) regret is possible, and (ii) the regime with log(T) regret is in fact interesting on its own. The latter is because:
>
> - problems with d=2 resources and small K arise in several motivating applications (see Appendix A) and capture the main challenges of BwK discussed in the Intro (Lines 29-35),
> - best-arm optimality is a typical, non-degenerate case (as we spell out in Footnote 5).
>
> Please note that our results on simple regret and on the general reduction are of independent interest. All of them are achieved with the same algorithm, so we are not sacrificing one type of guarantee for the sake of another. Our analyses introduce several new concepts & techniques (Lagrange gap, LP-gap, confidence sums, LP-sensitivity arguments for BwK).
>
> Please note that we discuss significance in detail in Section 6.
>
> ### Lack of logarithmic lower bounds, "whether $G_{\mbox{LAG}$ is a good or tight enough gap"
>
> - We rely on the logarithmic lower bound from bandits (Lai-Robbins [44]), which applies since Lagrange gap generalizes "reward-gap". In particular, 1/gap scaling is optimal. We point this out in Lines 137,139.
>
> - Moreover, our lower bounds in Section 3.2 show that sqrt{T} regret is inevitable for _some_ problem instances that have resources (i.e., do not reduce to bandits) and small Lagrange gap.
>
> - Lagrange gap is always well-defined and satisfies the desiderata of separating the dependence on T from the dependence on the (rest of) the problem instance. Formally, fixing the B/T ratio, Lagrange gap does not depend on T.
>
> For these reasons, we believe Lagrange gap is _a_ reasonable notion of "gap".
>
> However, Lagrange gap does not capture _all_ instances that admit O(log T) regret. This point should not be surprising per se, because neither does the standard notion of "reward-gap" for bandits! For example, problem instances with small reward-gap admit O(log T) regret if they have a likewise small best reward, e.g., via Thompson Sampling. So, perhaps there are "better" and/or incomparable notions of gap for BwK.
>
> A logarithmic lower bound for each instance of BwK would be very nice indeed! Any such bound would need to include some extra parameters and/or assumptions. (This point holds even for bandits and the "reward-gap": e.g., the bandit lower bound does not hold for instances with very small best reward.)
>
> We will include this discussion in the revision.
>
> ### "This paper is not quire well written as there are too many notations"
>
> BwK is inherently notation-heavy, and we use standard notation wherever possible (arms $a$, rewards $r$, consumption $c$, rounds $t$, etc etc.)
>
> Our non-standard notation highlights key concepts (various benchmarks, notions of "gap", LP-value, confidence sums, etc.) and hides unessential details (i.e., no need to "unwrap" these concepts many times inside derivations). Such notation is deliberately self-explanatory: e.g., Rad for "confidence radius", G_{LAG} for "Lagrange gap".
>
> FWIW, we've put much effort in polishing this paper, and we'll sure put in some more, and Rev1 (Rev "Ueua") thinks the paper is "nicely written". Perhaps some of the differences are a matter of personal taste.
>
> We'd appreciate concrete suggestions to help us revise.

---

> > ### Comment · Reviewer_9Bz5 · 2021-08-21
> > **Still having concerns, not changing the score**
> >
> > Thanks for the response. Although the result is limited and only works under a specific case, but I tend not to criticize the significance of the work. The novelty of the work is okay but it would be better if the authors can propose new ideas or algorithms. My main concern is the result, i.e., whether the value $G_{\mbox{LAG}}$ is good for measuring the gap. Can we find a way to have some basic results on the lower bound? Is there any difficulty stopping this?
> >
> > It is still unjustified that $G_{\mbox{LAG}}$ is a tight enough (or good enough) value for measuring the regret. I will expect some results on the lower bounds. Therefore, I cannot vote an accept but only marginal accept and maintain my score.

---

> > > ### Author Response · Authors · 2021-08-23
> > > **a response**
> > >
> > > Dear reviewer,
> > >
> > > **Re $G_{LAG}$:**
> > > In our response upthread, we list three reasons why we think $G_{LAG}$ is a reasonable notion of gap. Are you saying you don’t find them sufficient? In particular, there already is a "basic" lower bound which applies -- the one from Lai-Robbins.
> > >
> > > Presumably, what you are asking for is an instance-dependent lower bound which applies for every BwK instance, not just for special cases (of no resources and small $G_{LAG}$). We'd like to point out that **no such lower bound exists in prior work on BwK, for any notion of “gap”**. But we agree it would be nice to have one! One specific challenge, as we point out up-thread, is that such lower bound must include some additional parameters, just like Lai-Robbins cannot depend on the reward-gap alone.
> > >
> > > **Re novelty:**
> > > we’d like to re-iterate that we introduce several new concepts (Lagrange gap, LP-gap and confidence sums), several new techniques in the analysis, and the realization that UCBBwK leads to several types of regret bounds and these regret bounds easily generalize.
> > >
> > > Ironically, an earlier version of our algorithm required some crucial extra steps on top of what UCBBwK does; but then we figured out how to make the analysis work without these steps.
> > >
> > > In any case, thanks a lot for reviewing our paper!

---

### Official Review · Reviewer_Ueua · 2021-07-17

**Rating:** 6
**Confidence:** 3

**Summary:**

This paper tackles the Bandits with Knapsacks (BwK) problems, where the learner is not just looking for the arm with per-round maximal expected reward, but has to take into account the consumption of ressources that can end prematurely the whole process.
While existing work on BwK have already characterized the worst-case regret scaling in sqrt(T), this paper provides additional theoretical guarantees: problem depend upper bounds in log(T) for a UCB-like algorithm called UcbBwK, lower bounds matching the worst-case rate, as well as guarantees on the simple regret. Finally, when the problem instance presents some specific structure (e.g. in combinatorial semi-bandit, linear contextual bandits, multinomial-logit bandits), all these bounds for UcbBwK are improved.

**Limitations And Societal Impact:**



**Main Review:**

- The paper is nicely written and contains extensive discussion on the related literature.
- Strong theoretical guarantees are provided for the BwK problem.
- Unfortunately, no numerical experiments are provided to illustrate some mathematical claims. For instance, it would be interesting to verify empirically the relation between the parameter " beta " (line 326), the number of arms K, and the regret. In practice, how large can beta be?

**Time Spent Reviewing:**

1

---

> ### Author Response · Authors · 2021-08-10
> **Thanks for the positive comments on the paper.**
>
> Thanks for the positive comments on the paper.
>
> ###### Re lack of numerical experiments:
>
> This is primarily a theoretical paper, like most/all papers on BwK.  Besides, we do not propose a new algorithm, but provide tighter analyses of an existing one. So, while numerical experiments would be a nice complement for our theoretical results, we believe they would not add much extra value to the main message of the paper.
>
> ###### Re your specific question about the "beta" parameter in Section 5:
>
> The purpose of this result is to permit various extensions, as discussed in para 2 of Section 5. For each application, we identify a sufficient "beta", defined in terms of application-specific parameters, and immediately obtain several regret bounds.  Please note that the details are spelled out on Appendix H due to the page limit.  In particular, "beta" is not a free parameter in the algorithm.
>
> **Lack of experiments being the only criticism, we wonder if it justifies the "marginal accept".**

---

### Decision · Program_Chairs · 2021-09-27

**Decision:**

Accept (Poster)

**Comment:**

This paper investigates the bandits with knapsack (BwK) problem. It extends the state of the art by looking beyond the worst case regret analysis and provides logarithmic regret upper and lower bounds, first of their kinds. The second contribution of the paper is the analysis of BwK from the simple regret's perspective. Here the authors have proved that for any $\epsilon > 0$, apart from $O(K/\epsilon^2)\log(KTd))$ time steps, this simple regret is typically smaller than $\epsilon$. The final main contribution of the paper is the new proving concept that helps converting proof techniques from the classical bandit to the BwK setting, making the regret analysis in the BwK domain more seamless. Apart from this, the paper also introduces a number of new gap concepts and proof techniques.

As someone quite familiar with the BwK literature, I find this paper to be fascinating. If any weaknesses I need to mention then perhaps the denseness of the paper. It was a bit difficult to follow all the new concepts and terms. However, overall I find the amount of novelty and new ideas compensate this weakness.

From the reviewers' comments I can conclude that they also (more or less) share my judgement that this is a good paper. It is quite unfortunate that the reviewers were not involved in the discussion with the authors in a more active way to further elaborate on the strengths and weaknesses of the paper. Nevertheless, none of the reviewers did mention any reasons for rejecting this paper. Therefore, I recommend it to be accepted as a poster.